

# Using Multi-Criteria Decision Analysis for transdisciplinary co-design of the FANFAR flood forecasting and alert system in West Africa

Judit Lienert[1], Jafet Andersson[2], Daniel Hofmann[1], Francisco Silva Pinto[1], Martijn Kuller[1]

[1]Eawag: Swiss Federal Institute of Aquatic Science & Technology, Environmental Social Sciences (ESS) Department, Ueberlandstrasse 133, 8600 Duebendorf, Switzerland
[2]Swedish Meteorological and Hydrological Institute (SMHI), Hydrology Research, 601 76 Norrköping, Sweden

*Correspondence to*: Judit Lienert (judit.lienert@eawag.ch)

**Abstract.** Climate change is projected to increase flood risks in West Africa. The EU Horizon 2020 project FANFAR co-designed a pre-operational flood forecasting and alert system for West Africa in three lively workshops with 50–60 stakeholders, adopting a transdisciplinary framework from Multi-Criteria Decision Analysis (MCDA). We aimed to (i) exemplify MCDA as a structured transdisciplinary process; (ii) prioritize suitable FANFAR system configurations; and (iii) document and discuss empirical evidence. We used various interactive problem structuring methods in stakeholder sessions to generate 10 objectives and design 11 FANFAR system configurations. The non-additive MCDA model combined expert predictions about system performance with stakeholder preferences elicited in group sessions. All groups preferred a system producing accurate, clear, and accessible flood risk information that reaches recipients well before floods. To receive this, most groups would trade off higher operation and maintenance costs, development time, and implementing several languages. We accounted for uncertainty in expert predictions with Monte Carlo simulation. Sensitivity analyses tested the results' robustness for changing MCDA aggregation models and diverging stakeholder preferences. Despite many uncertainties, three FANFAR system configurations achieved 63–70 % of the ideal case over all objectives in all stakeholder groups, and outperformed other options in cost-benefit visualizations. Stakeholders designed these best options to work reliably under difficult West African conditions rather than incorporating many advanced features. The current FANFAR system combines important features increasing system performance. Most respondents of a small online survey are satisfied, and willing to use the system in future. We discuss our learning drawing from design principles of transdisciplinary research. We attempted to overcome "unbalanced ownership" and "insufficient legitimacy" by including key West African institutions as consortium partners and carrying out co-design workshops with mandated representatives from 17 countries. MCDA overcomes challenges such as "lack of technical integration", or "vagueness and ambiguity of results". Whether FANFAR will have a "societal impact" depends on long term financing and system uptake by West African institutions after termination of EU sponsoring. We hope that our promising results will have a "scientific impact" and motivate further stakeholder engagement in hydrology research.



# 1    Introduction

West Africa is vulnerable to the projected impacts of climate change, particularly related to runoff quantities (Aich et al., 2016;Roudier et al., 2014;Sylla et al., 2018). While the mechanisms and projections remain uncertain for West Africa, there is growing evidence for increased frequency, magnitude, and impact of fluvial floods (Nka et al., 2015;Ntajal et al., 2017). Already today, West Africa is heavily impacted by floods. Preliminary data from the United Nations estimate that 465 people died from floods in West and Central Africa in 2020. More than 1.7 million people were affected (double the number of the

year before), 94'000 people were displaced, and 152'000 houses were destroyed (OCHA, 2020). Worldwide, good operational flood forecast systems, giving accurate, timely, precise, and understandable forecast information and alerts, provide effective and affordable help with anticipating and minimizing impacts of flood events (Perera et al., 2019). Several systems have been set up for different West African regions, some being very useful. However, none seem to sufficiently meet stakeholder needs in terms of: i) timeliness (e.g., the annual frequency of the PRESASS and PRESAGG seasonal forecasts; WMO, 2021); ii)

coverage (e.g., systems based on propagation of streamflow measurements such as SLAPIDS, OPIDIN, FEWS-Oti often only cover a small part of West Africa and do not predict in ungauged basins; Massazza et al., 2020); iii) up-to-date operational production, because many models are used only for research (e.g., Aich et al., 2016;Schuol et al., 2008), and sometimes systems fail due to e.g., interrupted data flows or server failures (e.g., SATH-NBA in the major floods of 2020; NBA, 2021); iv) accuracy (e.g., global modelling systems such as FloFAS; Passerotti et al., 2020); and v) openness and ownership (e.g., pro-

prietary closed source consultancy systems, which may limit the capacity and independence of West African scientists an practitioners, and hence the long-term sustainability of such systems).

The EU Horizon 2020 project FANFAR, running from 2018 to 2021, was set up to address these gaps (FANFAR, 2021). FANFAR establishes and reinforces existing cooperation between European and West African hydrologists, information and communication technology experts, decision analysts, and end users such as West African hydrologists and emergency man-

agers (Andersson et al., 2020a). The aim of FANFAR is to co-design and co-develop a pre-operational flood forecasting and alert system at West Africa scale ("FANFAR system"). The FANFAR system is currently based on three open-source hydrological models, Niger-HYPE, West Africa HYPE, and World-Wide HYPE (Andersson et al., 2017;Arheimer et al., 2020), employed in a cloud-based ICT environment. The daily forecasting chain includes meteorological reanalysis and forecasting, data assimilation of gauge observations and satellite altimetry, hydrological initialization and forecasting, flood alert deriva-

tion, and distribution through Email, SMS, API, and a web based Interactive Visualization Portal (IVP, https://fanfar.eu/ivp/). In this paper, we do not address the technical system and refer interested readers to Andersson et al. (2020b). Rather, we emphasize the complex development of the FANFAR system in an iterative co-design process, the necessity of which has been recently underlined by Sultan et al. (2020). At its core are three lively one-week workshops carried out in West Africa from 2018–2020, each with 50–60 participants. We organized stakeholder participation adopting a transdisciplinary framework

from Multi-Criteria Decision Analysis (MCDA; Eisenführ et al., 2010;Keeney, 1982;Keeney and Raiffa, 1976;Belton and Stewart, 2002). Using MCDA, we integrated the stakeholders' preferences in the assessment of how well different FANFAR





system configurations meet stakeholder objectives. This helped to focus system development on configurations that best meet expectations, despite there being contradictory interests concerning the importance of objectives.

## 1.1 Aims and structure of paper

Our aims are: (i) to exemplify the use of Multi-Criteria Decision Analysis (MCDA) as methodological framework for integrating stakeholders in a structured co-design process; (ii) to prioritize development of suitable FANFAR flood forecast and alert system configurations based on expert estimates about system performance as well as stakeholder preferences; and (iii) to document empirical evidence of a large transdisciplinary, transcontinental co-design process, and discuss insights, lessons learnt, and recommendations of special interest to hydrology praxis when engaging with stakeholders and society.

The remainder of this paper is organized as follows: We shortly introduce transdisciplinary research (sect. 1.2), then motivate the use of a MCDA framework to structure stakeholder interactions (sect. 1.3). In the Methods (sect. 2), we introduce the co-design process based on workshops in West Africa, followed by an overview of the methods applied in each step of the MCDA process. In the Results (sect. 3), we present main results of the problem structuring steps of the MCDA including stakeholder analysis, the final MCDA results, and insights from sensitivity analyses to test the robustness of best performing FANFAR

system configurations under changing model assumptions. In the Discussion (sect. 4), we discuss the MCDA results in view of implications for the FANFAR aim of co-developing "A good pre-operational flood forecasting and alert system" for West Africa, hereby focusing on the problem structuring phases (sect. 4.1.1) and on dealing with uncertainty (sect. 4.1.2). We then discuss challenges of the transdisciplinary process and lessons learnt (sect. 4.2), including recommendations that may be useful for other transdisciplinary research projects, e.g., in hydrology. We finish with some concluding remarks (sect. 5).

## 1.2 Transdisciplinary research

The close collaboration and partnership envisaged in the FANFAR project is ambitious. It responds to a growing awareness in disaster management that the development of early warning systems should closely involve end users to increase the systems' usefulness (adapted to end users' needs and preferences), and effectiveness (e.g., enhancing uptake of the system; Basher, 2006;Bierens et al., 2020;UNISDR, 2010). Such collaborations require high level inter- and transdisciplinary interactions,

which we shortly clarify here. Definitions of transdisciplinarity might be somewhat unclear in scholarly debates (Russell et al., 2008), but usually include collaboration between different scientific disciplines and "cooperation between researchers and practitioners" (Jahn et al., 2012). Our approach is captured by a definition for sustainability science: "Transdisciplinarity is a reflexive, integrative, method-driven scientific principle aiming at the solution or transition of societal problems and concurrently of related scientific problems by differentiating and integrating knowledge from various scientific and societal bodies

of knowledge" (Lang et al., 2012). The FANFAR project is unique in that it operates in a transnational and transcontinental context, with consortium partners across Europe (Italy, Spain, Sweden, and Switzerland) and Africa (from the organizations





AGRHYMET and NIHSA), and including stakeholders across 17 countries of West and Central Africa. Challenges of "ordi-nary" transdisciplinary research projects, which are often substantial, may become more accentuated in a transnational or transcontinental context. Lang et al. (2012) discuss some of these challenges along with suggestions for coping strategies.

They provide various empirical examples from around the globe, including Africa (see Lang et al., 2012 for references to specific cases). We address some issues in the Discussion (sect. 4.2).

In this paper, we hope to contribute to the lively discourse in the transdisciplinary research field with evidence from a large and presumably unique practice and outcome oriented project. The success of FANFAR relies on intensive collaborative efforts between different scientific disciplines and stakeholders in West Africa that have the mandate to produce reliable forecasts

and issue warnings. According to the call of this Special Issue, "(after a scientific decade on change in hydrology and society), transdisciplinary endeavors remain largely undocumented" (Carr et al., 2021). One important aim of this paper is to document our experience and lessons learnt, hereby contributing to knowledge production, learning, and scientific praxis in hydrology.

### 1.3    Multi-Criteria Decision Analysis (MCDA)

Although methodological and epistemological perspectives in transdisciplinary research may still be debated (Jahn et al.,

2012), there clearly needs to be consensus on methods used in a project, and on the concept to integrate research results (Lang et al., 2012). The FANFAR consortium agreed to use Multi-Criteria Decision Analysis (MCDA) as organizing framework to integrate the stakeholders in the transdisciplinary co-design process, and to achieve the overall aim of producing a co-designed system. MCDA embraces various specific methodologies to support complex decisions (e.g., Belton and Stewart, 2002).

We chose MCDA, and specifically Multi-Attribute Value Theory (MAVT; Eisenführ et al., 2010;Keeney, 1982;Keeney and

Raiffa, 1976) for several reasons: (i) developing a complex forecast system is not straightforward, and many decisions have to be made. As examples, the developers need to clarify, which data should be used to produce flood forecasts (e.g., type of meteorological forecast data and streamflow gauge observations), and which hydrological models. They need to decide, which flood hazard thresholds are appropriate, how model forecasts are visualized, or which distribution channels effectively reach people. MCDA is optimally suited to clarify such questions. (ii) Because the FANFAR system should be adapted to the needs

and preferences of stakeholders, its development relies on close collaboration with nonacademic practice partners. MCDA is a proven methodology that allows close interaction with stakeholders at various stages of decision making processes, with a choice of methods available for each stage. (iii) We use MAVT because it bases decisions on the values and objectives that are of fundamental importance to stakeholders. (iv) To evaluate options, MCDA allows comparing very different kinds of data, and integrating facts such as scientific and technical data from expert predictions (e.g., accuracy of forecasts, estimated

costs for system development) with stakeholders' preferences. Disentangling facts from values can be very helpful, especially if stakeholders have conflicting interests (Gregory et al., 2012a;Keeney, 1982). Tradeoffs have to be made in any complex decision where not all objectives can be fully achieved. MCDA explicitly asks stakeholders for the tradeoffs they are willing to make. (v) MCDA allows including various types of uncertainty, e.g., the uncertainty of scientific and expert predictions





with probability theory, or uncertain stakeholder preferences with sensitivity analyses (Reichert et al., 2015). (vi) MCDA is
carried out stepwise, splitting the decision into manageable parts. This reduces complexity and increases transparency.

For these reasons, we used MCDA to identify the configuration of a "Good flood forecast and alert system" for West Africa,
the "overall objective" in our MCDA. We present results of this typical transdisciplinary decision making process, including
discussing how we were able to identify system configurations that meet the various, sometimes conflicting stakeholder ex-
pectations. Additionally, we attempt to present the MCDA methods such that they are easily accessible and adaptable to other
transdisciplinary projects, e.g., in hydrology. Extensive details are provided as blueprint in the Supplementary Information.

## 2   Methods

### 2.1   Transdisciplinary co-design process using MCDA

Transdisciplinary research proposes a conceptual co-design framework consisting of three phases: (i) forming a common re-
search object (Jahn et al., 2012), or problem framing and team building (Lang et al., 2012), (ii) co-creating solution oriented
and transferable knowledge through collaborative research, (iii) applying the co-produced knowledge (Lang et al., 2012), and
evaluating its contribution to societal and scientific progress (Jahn et al., 2012). This process is iterative, which is captured in
the FANFAR co-design process with several cycles of meeting with decision makers, end users, and stakeholders (referred to
as stakeholders hereafter) to test, discuss, and improve the FANFAR pre-operational system.

We start the MCDA process with problem framing (Figure 1), as in phase one of the transdisciplinary model. This entails
properly defining the decision. We undertook a separate stakeholder analysis (e.g., Grimble and Wellard, 1997;Lienert et al.,
2013;Reed et al., 2009), which is not always part of MCDA. This was especially important, since European researchers are
working in an unfamiliar African context. Main identified stakeholders that participated in the workshops were representatives
from hydrological services, emergency management agencies, river basin organizations, and regional expert agencies. To-
gether with these priority stakeholders, we then identified objectives ("What is of fundamental importance to be achieved by
a FANFAR system?") and options ("Which FANFAR system configurations are potentially suitable to achieve objectives?").
In MCDA, these first steps are summarized as "problem structuring", and a diverse choice of "Problem Structuring Methods"
(PSMs) is available in other fields of Decision Analysis (Rosenhead and Mingers, 2001). It is common to combine MCDA
with selected PSMs (reviewed by Marttunen et al., 2017). For a detailed description of similar PSMs as used in FANFAR, we
refer to an application in wastewater infrastructure planning (Lienert et al., 2015). The next steps 5–7 in MCDA (Figure 1),
belong to phase two of the transdisciplinary model (Jahn et al., 2012;Lang et al., 2012). In phase three, this co-created new
knowledge is evaluated and/or applied. Below, we summarize the co-design workshops (sect. 2.2), before focusing on each
step of the MCDA process (sect. 2.3 to sect. 2.9).



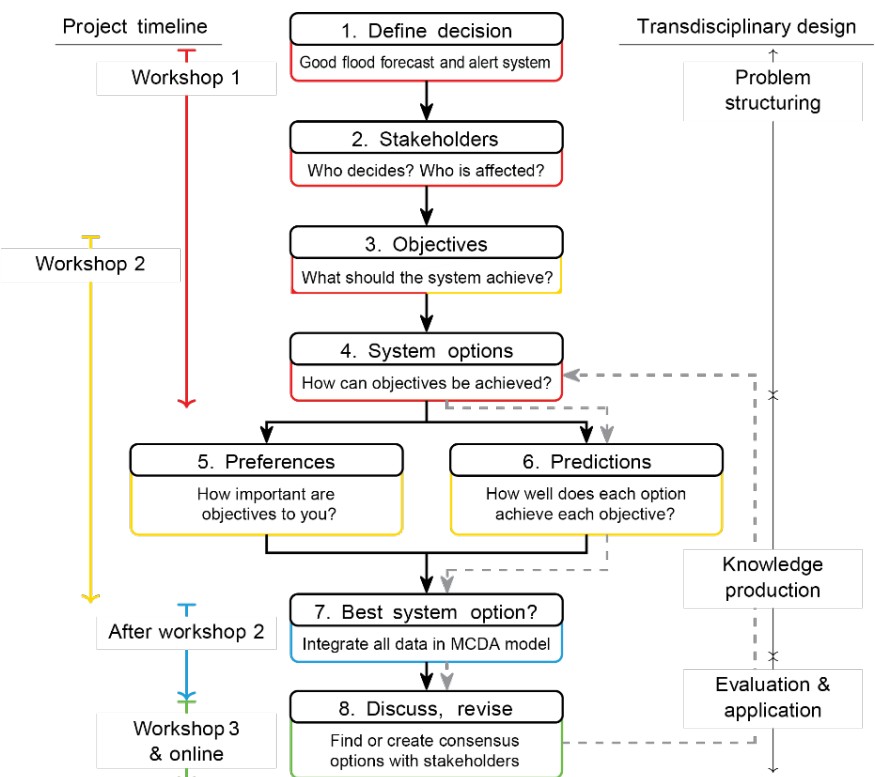

**Figure 1: Multi-Criteria Decision Analysis (MCDA) is carried out stepwise in the FANFAR project. Explanations see text.**

## 2.2 Co-design workshops in West Africa

To date (March 2021), three co-design workshops have been carried out in West Africa, and a kick off meeting of the FANFAR consortium in Norrköping (Sweden, 17–18 January 2018). A fourth workshop in West Africa had to be replaced by two half-day online workshops due to COVID-19 (20–21 January 2021). The co-design process is documented in a detailed report (Lienert et al., 2020), and workshop summary reports (FANFAR, 2021). At each workshop, West African representatives presented the local flood situation in their country during rainy seasons and their experience with using the FANFAR system. Each workshop hosted extensive technical sessions for experimentation with the latest FANFAR system, including structured technical feedback. Between workshops, the pre-operational system was adapted to meet requests as well as possible (Andersson et al., 2020a). Furthermore, we conducted sessions with emergency managers, e.g., about their understanding of flood risk representation to improve visualizations of the FANFAR system (Kuller et al., 2020). In this paper, we focus only on interactions that are at the core of the MCDA process.

The first co-design workshop in Niamey (Niger, 17–20 September 2018) hosted 47 participants from 21 countries, including consortium members from Europe and Africa, and representatives from hydrological service agencies and emergency management agencies on regional and national levels from 17 countries in West and Central Africa. Main aim was to initiate the





co-design process. For the MCDA, we carried out the problem structuring steps (Figure 1): a stakeholder analysis (sect. 2.3);
different interactions to identify fundamentally important objectives of participants (i.e., what the system should achieve; sect.
2.4), and to identify options (i.e., how to configure the system to meet objectives; sect. 2.5). The second co-design workshop
in Accra (Ghana, 9–12 April 2019) hosted 48 participants from 21 countries. For the MCDA, we consolidated the list of
objectives, and elicited participants' preferences regarding achieving these objectives in small groups (sect. 2.7). Additionally,
we collected preference data on the importance of objectives from each individual participant with questionnaires. This gave
interesting insights concerning preference formation and changes over time (Kuller et al., in prep.). In the third workshop in
Abuja (Nigeria, 10–14 February 2020), the number of participants increased to 58, including representatives from the World
Meteorological Organization (WMO; https://public.wmo.int/) and the Economic Community of West African States (ECO-
WAS; https://www.ecowas.int/), and representatives from 16 West and Central African countries. We discussed main MCDA
results. During the fourth online workshop, stakeholders completed a survey, providing some feedback for the MCDA.

## 2.3 Stakeholder analysis

For the stakeholder analysis (Grimble and Wellard, 1997;Reed et al., 2009), we followed the procedure in Lienert et al. (2013).
The workshop participants filled in a pen and paper questionnaire in French or English, assisted by two experts. The survey
was completed in 2.5 hours by 18 groups consisting of two to three people from the same country, with a total of 31 participants.
After receiving information and instructions, the participants completed two tables, one for identifying key West African or-
ganizations involved in producing and operating flood forecast and early warning systems, and one for identifying downstream
stakeholders (i.e., "Who might play a role because they use information from such systems in society?"). Each table contained
eight tasks: (1) listing key organizations or stakeholders; (2) specifications (e.g., representative names); (3) their (presumed)
main interests; (4) why they might use the FANFAR system; and (5) which distribution channels are appropriate. We then
used a 10 point Likert scale, asking the participants to (6) rate the importance of considering each listed stakeholder or organ-
ization in the FANFAR co-design process; (7) the presumed influence (power) of each stakeholder in the implementation of
the FANFAR system; and (8) how strongly each stakeholder or organization would be affected by the system (i.e., its level of
performance). We cleaned the raw data and categorized stakeholders based on whether they are forecast/alert producers or end
users, their decisional level, sector, and perceived main interest. More details see Silva Pinto and Lienert (2018).

## 2.4 Generating objectives and attributes

Generating objectives is key to MCDA (Belton and Stewart, 2002;Eisenführ et al., 2010;Keeney, 1982;Keeney and Raiffa,
1976), since this choice can significantly alter results. Simply asking stakeholders is insufficient, and often too few (Bond et
al., 2008;Haag et al., 2019c) or too many objectives are produced; we refer to the useful guidelines in Marttunen et al. (2019).
Our stepwise procedure started at the FANFAR kick off meeting in Sweden and continued in the first two West African co-
design workshops (details see Lienert et al., 2020). In the first workshop, participants worked in three parallel groups. Individ-
uals in the first group used an interactive online survey to first brainstorm, then select objectives from a master list (Haag et





al., 2019c). Individuals in the second group used the same procedure in a pen and paper survey assisted by a moderator. The third group used a means-ends network in a moderated group discussion to come up with consensus objectives (Eisenführ et al., 2010). Each participant (respectively group), ranked and rated objectives according to importance. Objectives were discussed in a lively plenary and the most important ones chosen by majority vote. Between workshops, we post processed
objectives to avoid common mistakes such as double counting and overlaps, or including means objectives (Eisenführ et al., 2010). MCDA objectives are only useful if they discriminate options, and we dismissed those not fulfilling this requirement, even if such objectives can be important to stakeholders. In the second workshop, we presented a revised list of the 10 most important objectives, including a clear definition and description of the best and worst possible case for each (see first part of detailed description for each attribute in sect. SI-2.4.1). For instance, for the objective "Several languages", the FANFAR
system being available in several languages is the best case, and only in English the worst. After discussion, the workshop participants agreed on the final list of objectives as basis for the MCDA. To operationalize objectives, attributes (synonym indicators) are required (Eisenführ et al., 2010). These were developed by experts from the FANFAR consortium. In most cases, we constructed attributes from several sub-attributes (sect. 2.6). Sub-attributes were transformed to a value with help of marginal value functions (sect. 2.7.1) to allow aggregation into a single value for the respective attribute. The attributes were
then aggregated into a single total value using the MCDA model (sect. 2.8).

## 2.5 Generating system options

Different plausible FANFAR system options were generated in the first co-design workshop, in three moderated group sessions. Two groups used the "Strategy Generation Table" (Gregory et al., 2012b;Howard, 1988), and one "Brainwriting 635" (Litcanu et al., 2015;Paulus and Yang, 2000) combined with "Cadavre Exquis" (words written on a paper that is folded, then
handed over to next person). The Strategy Generation Table is a systematic procedure that allows pre-structuring elements of the FANFAR system (e.g., observed variables, models for forecast production, language). The stakeholders chose elements forming suitable system configurations ("strategies") with help of questions such as: "The most easy to use system", or the "Most robust system working well given boundary conditions in West Africa (e.g., internet or power supply problems)". Brainwriting 635 allowed for more open brainstorming, with the aim to also interactively develop additional system options
using the same strategies as in the other sessions. All FANFAR system options were discussed in a plenary session. As part of post processing, additional technically interesting system options were created by FANFAR consortium members. For readers unfamiliar with the methods, we provide details in the Supplementary Information (sect. SI-1.1).

## 2.6 Predicting performance of each system option

Part of the input data for the MCDA model consists of scientific predictions (Figure 1), based on estimates or models of the
level of achievement for each objective (Eisenführ et al., 2010). We used expert estimates (O'Hagan, 2019) by interviewing FANFAR consortium members in July–August 2019. First, the experts developed attributes (sect. 2.4), in most cases constructed from sub-attributes. The experts then estimated the outcome of each FANFAR system option for each (sub-)attribute,



i.e., the most likely level of each attribute (e.g., likely operation costs). The experts also stated ranges of uncertainty regarding their predictions. For constructed attributes, we integrated the predictions of the sub-attributes into one final value using a

weighted sum, with weights defined by the experts (sect. 2.7). We aggregated the uncertainty of each sub-attribute into a single uncertainty distribution with 1'000 Monte Carlo simulations. To characterize the resulting aggregated uncertainty, we used a normal distribution with mean (of Monte Carlo simulation), and standard deviation (¼ of the 95 % confidence interval from simulation) as input in the subsequent MCDA with uncertainty (sect. 2.8).

We give one example of constructing attributes following MCDA principles: The objective "High accuracy of information"

consists of three sub-attributes, the KGE index for 1 day, 3 day, and 10 day forecasts (Kling-Gupta Efficiency; Gupta et al., 2009). The KGE is one of several possible accuracy indices for model evaluation in hydrology, e.g., to estimate the error of predicted vs. observed values. For each FANFAR system option and each lead day, the expert estimated the expected KGE. The KGE index number was then transformed to a value, ranging from 0 (worst) to 1 (best), using a nonlinear marginal value function, elicited from the expert. We aggregated the lead day values into a single value [0:1] with a weighted sum, where the

accuracy of the 1 day forecast received a weight of 0.5, the 3 day forecast of 0.4, and the 10 day forecast of 0.1. We give details for predicting system performance (i.e., how well it is expected to achieve stakeholder objectives) in sect. SI-2.4.

## 2.7   Eliciting stakeholder preferences

### 2.7.1   Marginal value functions

Subjective preferences of stakeholders enter the MCDA model on equal footing to the expert predictions (Figure 1). Preference

elicitation is an important, sensitive step during which many biases can occur (Montibeller and von Winterfeldt, 2015). It is thus crucial to follow recommendations (Eisenführ et al., 2010). Marginal value functions convert the attribute levels for each objective (e.g., KGE index for "High accuracy of information") to a common scale ranging from 0 (worst possible achievement of this objective) to 1 (best possible achievement). This transformation allows integrating different attributes with various units into one model, e.g., the KGE index with operation costs (€ / year), and development time (days). As default, a linear marginal

value function can be used. However, nonlinear value functions usually better capture preferences. In FANFAR, most attributes are relatively technical, requiring expert knowledge. We therefore elicited shapes of value functions from experts when making the predictions (sect. 2.6; details, including figures of value functions see sect. SI-2.4.1).

For each sub-attribute, we mostly created eight evenly spaced levels (worst, very bad, bad, neutral, good, very good, and best). Experts then assigned attribute numbers (e.g., KGE index for 3 day forecasts) to each level based on their experience. We

transformed attribute levels to [0:1] values using linear interpolation between levels. As example, the KGE index ranges from –infinity, which is the worst case receiving a value of 0, to 1, the best case with value 1 (Table SI-8). For attributes consisting of sub-attributes, we had elicited a nonlinear marginal value function for each sub-attribute (Figure SI-5), allowing aggregation into one single value. As consequence, because we had already used elicited nonlinear value functions to construct the composite attribute, we used a linear value function for constructed attributes in the subsequent MCDA modelling (sect. 2.8).



### 2.7.2    Weights

In the second FANFAR co-design workshop, we elicited the weights of the workshop participants in group sessions. We divided participants into five groups according to language (French F, English E) and professional background (Emergency Managers EM, Hydrologists HY). The two French speaking groups used the popular Swing method (Eisenführ et al., 2010): eight emergency managers (group ID: G1A_EM_F, where G1A = group), and 11 hydrologists (in two sub-groups, G2A_HY_F, G2B_HY_F). The two English speaking groups used an adaptation of Simos' revised card procedure (Figueira and Roy, 2002;Pictet and Bollinger, 2008), hereafter Simos card: 14 hydrologists (G3A_HY_E), and three emergency managers (G4A_EM_E). We separately elicited weights from three AGRHYMET experts with Simos card method (G5A_AGRHYMET). They are FANFAR consortium member with an important regional role and high hydrology expertise.

Stakeholders can be uncertain about their preferences or groups may disagree. For Swing, we avoided forcing the participants to reach group consensus, encouraging them to discuss diverging opinions. This resulted in a range of the stakeholders' weight preferences. We took the mean as main weight and considered strong deviations (difference in weights > 0.2 compared to mean) in later sensitivity analyses (sect. 2.9.2). Two additional weight sets resulted from Simos card procedure, because a range was elicited for one variable. The moderator recorded important comments to inform the sensitivity analyses (Table SI-3). In the group of French speaking hydrologists, two diverging preference sets emerged from the start, which we analyzed separately (G2A, G2B). For interested readers, we give details concerning these standard MCDA weight elicitation procedures in sect. SI-1.2. To check for the validity of the additive aggregation model (sect. 2.8), we shortly discussed implications in the weight elicitation sessions using elicitation procedures from our earlier work (Haag et al., 2019a;Zheng et al., 2016).

### 2.8    MCDA model integrating predictions and preferences

The MCDA model integrates the expert predictions with the stakeholders' preferences, which are captured as model preference parameters. An aggregation model is used to calculate the total value of each option (Eisenführ et al., 2010). Concretely, a finite set of alternatives (or options) $A = \{a, b, \dots\}$ are evaluated regarding the predicted outcomes on every objective (respectively attribute). We denote the predicted outcomes (sect. 2.6) as $x_a = (x_{a,1}, \dots, x_{a,n})$, with $x_{a,i}$ being the level of an attribute $i$ that measures a predicted consequence of option $a$ *(or b, c, …)*. The total value $v(x_a)$ of an option $a$ is calculated with a multi-attribute value function, $v(x_{a,1}, \dots, x_{a,n}, \theta)$. The resulting total value *v(xₐ)* of each option lies between 0 (all objectives achieve only the worst level) and 1 (all objectives are on the best attribute level that can be achieved given the defined attribute ranges). A rational decision maker would choose the option with the highest value. Most commonly, an additive model based on single-attribute value functions is used, as in Eq. (1) and Eq. (2), but non-additive models as in Eq. (3) are also possible:

$$v(x_1, x_2, \dots, x_n, \theta) = \sum_{i=1}^{n} w_i \cdot v_i(x_i, \theta) \qquad\qquad\qquad \text{(Eq. 1)}$$

with parameters $\theta = (w_1, \dots, w_n, \theta)$, where $w_i$ is the weight of attribute $i$ (see sect. 2.7.2), with $0 \le w_i \le 1$, and

$$\sum_{i=1}^{n} w_i = 1, \qquad\qquad\qquad\qquad\qquad\qquad\qquad\qquad\qquad \text{(Eq. 2)}$$





and where $v_i(x_i, \theta)$ is the value for the predicted consequence $x_i$ of attribute $i$ of option $a$ This value is inferred with help of the marginal value function (sect. 2.7.1).

While easy to understand, the additive model entails strong assumptions, e.g., that objectives are preferentially independent (Eisenführ et al., 2010). Increasing evidence indicates that many stakeholders do not agree with model implications (Haag et al., 2019a;Reichert et al., 2019;Zheng et al., 2016). Additive aggregation implies that good performance on one objective can fully compensate for poor performance on another. In the FANFAR weight elicitation sessions, we asked stakeholders, using some examples, whether they agree with objectives being preferentially independent, and as consequence with the full compensatory effect. In all five groups this was not the case. We used a non-additive model with less strict requirements, the weighted power mean with an additional parameter $\gamma$ that determines the degree of non-compensation, see Eq. (3):

$$v(x_1, x_2, \ldots, x_n, \theta) = \left(\sum_{i=1}^{n} w_i \cdot v_i(x_i, \theta)^{\gamma}\right)^{1/\gamma} \qquad \text{(Eq. 3)}$$

If $\gamma = 1$, we are back to the additive model in Eq. (1). We used a value for $\gamma = 0.2$, based on input from the stakeholders (sect. 2.7.2), which is closer to a weighted geometric mean ($\gamma \to 0$). We shortly explain and visualize the implications of the power mean in sect. SI-1.3. For further details we refer to (Haag et al., 2019b).

We calculated all MCDA results in our newly developed open source software application "ValueDecisions" (Haag et al., in prep.). ValueDecisions is based on the open source software and programming language R (Team, 2018), earlier R scripts developed in our group (e.g., Haag et al., 2019b), and the R "utility" package (Reichert et al., 2013). R scripts were rendered as web application for ValueDecisions with the "shiny" package (Chang et al., 2020). Additional analyses were implemented directly in R: aggregating uncertainty of sub-attributes, weight visualization, and statistical analysis of sensitivity analyses.

### 2.9    Uncertainty of predictions and preferences

#### 2.9.1    Uncertainty of predictions

To deal with the uncertainty of predictions, probability theory is commonly used in MAVT (Reichert et al., 2015). We defined uncertainty distributions for the expert predictions for each attribute as explained above (sect. 2.6). When calculating the aggregated values of the system options across all objectives with the MCDA model (sect. 2.8), we used 1'000 Monte Carlo simulation runs, drawing randomly from the attributes' uncertainty distributions. We then analyzed the rank frequencies of each system option, i.e., how many times of the 1'000 runs each system option achieved each rank.

#### 2.9.2    Sensitivity analyses of aggregation model and stakeholder preferences

Local sensitivity analyses are commonly used to check the sensitivity of the MCDA results to diverging preferences (e.g., Eisenführ et al., 2010;Zheng et al., 2016;Haag et al., in prep.). The MCDA is recalculated with changed preference parameters to identify whether the changed input provokes rank reversals and/or strong changes in the total values of options. We checked the sensitivity of our results to other aggregation models and to changed weights, since there was not always consensus in the





weight elicitation groups. Essentially, the weight of one objective is changed, while the ratios of all other weights are kept constant, and renormalized so that the sum of all weights remains 1. For a thorough method explanation we refer to Eisenführ et al. (2010) and give some insight for readers not familiar with MCDA in sect. SI-1.4.

To test implications of the aggregation model (sect. 2.8), we recalculated the MCDA for other reasonable models (Haag et al.,
2019a), running a separate MCDA for each. Applying alternative models requires one mouse click in the ValueDecisions app (Haag et al., in prep.). We give the settings of our sensitivity analyses in the Results (Table 2), with setting S0 as default. Consistency checks during weight elicitation with the French speaking emergency managers (G1A) revealed an inconsistency for the higher level objectives. It was not possible to resolve this during the workshop (Table SI-3). The differences between weights of these two sub-groups were large (Figure SI-3). We tested their effect on the results in sensitivity analysis S21 (Table
2). For Swing weight elicitation, we had allowed workshop participants to state ranges rather than precise numbers. In cases where the difference between the maximum or minimum weight from the average weight exceeded $\Delta = 0.02$, we first used weight sensitivity plots to visually investigate whether the respective objective (for the respective group) is sensitive towards weight changes (example see Figure SI-4). Thereafter, we recalculated the MCDA for each identified case, using the maximum, respectively minimum weight of the sensitive objective (S22). For Simos' card method, the Z value was elicited as range
(sect SI-1.2.2). We recalculated the MCDA for each group using the additional weight set resulting from the minimum Z value (S231). Using the maximum Z value was only necessary for group G3A (S232), because Z min was used as focus value for groups G4A and G5A (i.e., as S0). It is common to test other interesting objectives by doubling the elicited weight. We did this for "Several languages", because the importance might have been underestimated (S31). For each setting, we compared results (i.e., ranks of options) with those of the default MCDA (S0; Table 2), using the nonparametric Kendall's $\tau$ correlation
coefficient (Kendall, 1938) to measure rank reversals (as in e.g., Zheng et al., 2016). As input in the correlation analyses, we used the mean ranks resulting from 1'000 Monte Carlo simulation runs with the changed parameter settings.

Cost-benefit visualizations in the ValueDecisions app are an additional way to check the robustness of options (e.g., Liu et al., 2019). This visual analysis is based on the resulting MCDA, including the stakeholder preferences. For each group, we used standard setting S0 without uncertainty of predictions (Table 2). First, we plotted the values of the total operation and mainte-
nance costs (i.e., attribute *32_costs*) of each FANFAR system option against all other "benefits". The benefits were the total aggregated value that each option achieved for the other nine objectives. The options with highest performance on both the x- and y-axis were then linearly connected. This interpolation visualizes a possible efficient frontier for the given options, and any option below this line is outperformed by one or more other options in this set. We repeated the visualization for the aggregated value of *32_costs* and the time needed to develop the FANFAR system (*31_develop_time*) as costs, plotted against
the total aggregated value of all other eight objectives. Third, we plotted the aggregated value of *32_costs*, *31_develop_time*, and secured long-term financing (*41_sust_financing*) as costs against the total aggregated value of all other seven objectives.



### 2.10 Discuss results with stakeholders, feedback

We presented and discussed preliminary MCDA results with stakeholders in the third co-design workshop. Because the fourth workshop had to be carried out online, we were not able to thoroughly discuss results with the stakeholders. However, we did

assess stakeholder perceived satisfaction with the performance of the FANFAR system during the 2020 rainy season using an online survey to answer following questions for each objective: (a) How much does FANFAR currently fulfill this objective? (b) Would you use the FANFAR system in the future if it remains as is? (c) What is the minimum acceptable to you? This means: below which level would you NOT use the FANFAR system? (details see sect. SI-1.5). The survey was filled out by 12 participants, resulting for our 10 objectives in 10 x 12 = 120 responses to each question.

## 3 Results

### 3.1 Stakeholder analysis

Of 249 stakeholders listed by workshop participants, 68 distinct types remained after data cleaning (details see Silva Pinto and Lienert, 2018). Stakeholders that were perceived to have high influence and also potentially being highly affected by the FANFAR system were national entities for disaster management planning and water resources and infrastructure, who were

already well represented in the co-design process (details Table SI-4). Several specific organizations were also perceived as both highly important and strongly affected, such as the "Autorité du Bassin de la Volta" (ABV), who participated in the workshops, and AGRHYMET, an organization representing 13 West African states and consortium member. Other important/affected parties were mainly stakeholders receiving flood forecasts and alerts such as NGO's, electricity utilities, dam managers, and the agricultural sector. These groups are reflected in estimated prospective usage numbers of the FANFAR

system as perceived by the respondents: half of the stakeholders (46 %) would likely use the FANFAR system for "alert information", 16 % for "water related information", and 21 % for "forecast refinement". Only 4 % would use it for own "forecast production". The Red Cross and environmental protection agencies were perceived to have slightly lower importance/affectedness, among others. Civil society (e.g., communities) would be strongly affected, but have limited decisional influence on developing the FANFAR system, and decision making in general. In contrast, the media, industry, and commerce

were perceived to have more influence, but would not be strongly affected. Such outlier stakeholders could potentially provide a different view to co-designing the FANFAR system.

### 3.2 Objectives and attributes

The objectives generated by stakeholders covered a broad range of issues that they considered as fundamentally important in view of a *Good flood forecast and alert system* for West Africa (all objectives, attributes, and options are given in Italics

hereafter). We grouped these into an objectives hierarchy (Figure 2). Several objectives concerned quality requirements for



the forecasted flood information, grouped under *High information accuracy and clarity,* and different aspects of *Good information access* by users such as accounting for the language diversity in the region. Aspects of *Low costs* and on a longer term *High sustainability* were also important, such as *Skillful human resources available* in West Africa, capable of maintaining, operating, and accessing the system. Each objective is characterized by an attribute in MCDA, which allows operationalizing

the achievement of objectives (Figure 2). Details concerning attribute calculations are given in sect. SI-2.4.

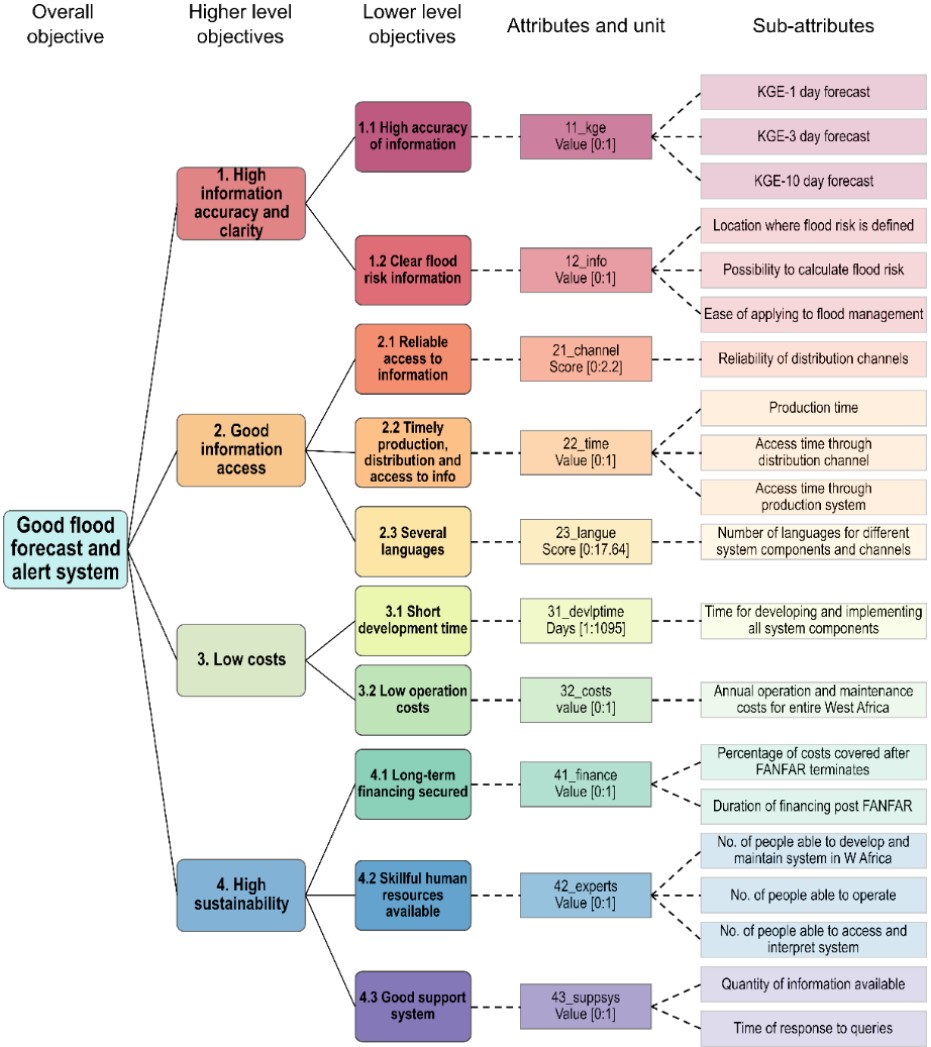

**Figure 2. Objectives hierarchy. From left to right: overall objective, four higher level fundamental objectives, 10 lower level fundamental objectives and corresponding attributes, attributes' unit (usually a value) and range [square brackets], from worst (usually value = 0) to best (usually value = 1). Most attributes were constructed from several sub-attributes (far-right).**


**3.3    FANFAR system options**

Stakeholders generated six system options (options b to g; Table 1) in workshop sessions. Experts of the FANFAR consortium developed five additional options to cover important technical aspects such as using more refined hydrological models, e.g., redelineation and recalibration of the World Wide HYPE (WWH) model to West Africa (Andersson et al., 2020b), and includ-ing earth observation data from satellites (options h to k). Options were constructed in separate sessions with experts from

AGRHYMET for the forecast production system, and the user interface IVP (Interactive Visualization Portal) with stakehold-ers. They were combined to form plausible combinations of various FANFAR system elements (summary of important features in Table 1; for all system elements see Table SI-6 and Table SI-7). Option *a_Fast-dev* represents roughly the state of the first version of the FANFAR system, when stakeholders started experimentation and giving feedback in the first workshop.

**Table 1. Overview of 11 FANFAR system options. Selected main characteristics: recent hydrological observation data types**
**(HydObs; WL: water level, Q: river discharge, EO: Earth Observations) & meteorological input/forcing data (MetF; HydroGFD2 (Berg et al., 2018); HydroGFD3 (Berg et al., 2020; improved version); HydroGFD-WA: HydroGFD2 adjusted by West African meteorological observations; Am: American meteorological forecasts (e.g., GFS); Ens: ECMWF ensemble meteorological forecasts); hydrological models (WWH: World-Wide HYPE); forecast output variables (Q: river discharge; WL: water level, P: precipitation; E: evaporation; SM: soil moisture, WQ: water quality); data download (Excel: table for selected station); distribution channels**
**(Web: web visualization; H-TEP: login to H-TEP to download data; FTP: FANFAR and national FTP; API: Application Program-ming Interface; SoMed: Social Media e.g., WhatsApp; ConMed: conventional media e.g., radio, TV; Tradit: traditional word of mouth) & automatization (Automatic: automatic push of data to distribution channels; Manual: automatic processing with manual control of distribution by operator); flood hazard reference threshold types (RP Sim: return period based on simulations; RP Obs: return periods based on observations at gauged locations; HistY: selected historic year; Local: user defined thresholds for specific**
**location); language of user interface (En: English; Fr: French; Pt: Portuguese; Ar: Arabic).**

| ID | Option | Hydrological observations & meteorological forcing | Hydrological models | Forecast output variables | Data download | Distribution channels & automatization | Flood hazard thresholds | Language |
|---|---|---|---|---|---|---|---|---|
| a_Fast-dev | Least resources for de-velopment: no new fea-tures, status quo | HydObs: none; MetF: Hy-droGFD2 | Niger HYPE | Q | None | Web; Automatic | RP Sim | En |
| b_Res-user | Least resources for us-ers (e.g., skilled person-nel, stable internet and power) | HydObs: in situ WL, Q; MetF: HydroGFD3 | WWH | Q, WL, P, E, SM | Excel, maps, graphs | Web, H-TEP, SMS, Email, SoMed, Con-Med, Tradit; Manual | RP Sim, RP Obs, HistY, Local | En, Fr, Pt, Ar |
| c_Easy-use | Most easy to use for producing and interpret-ing forecasts and alerts | HydObs: EO WL; MetF: Hy-droGFD2 | Niger HYPE | Q, WL, P, E | Excel, graphs | Web, SMS, SoMed, ConMed, Tradit; Au-tomatic | RP Sim, HistY | En, Fr, Pt |





| | | | | | | | | |
|---|---|---|---|---|---|---|---|---|
| d_Fast | Fastest system for producing and distributing forecasts and alerts | HydObs: EO WL; MetF: HydroGFD2 | Niger HYPE | Q | None | Web, SMS, Email, SoMed, ConMed, Tradit; Automatic | RP Sim | En |
| e_Consent | Highest consensus: system elements that West African stakeholders mostly agreed on | HydObs: in situ WL, Q, EO WL; MetF: HydroGFD-WA, Am, Ens | Niger HYPE, WWH | Q, WL, P, E, SM | Excel, maps, graphs | Web, H-TEP, SMS, Email, SoMed, ConMed, Tradit; Manual | RP Sim, HistY | En, Fr, Pt |
| f_Robust | Most robust in West Africa: works despite problems in e.g., data collection | HydObs: EO WL; MetF: HydroGFD2 | Niger HYPE, WWH | Q, WL, P, E, SM | Excel, maps, graphs | Web, H-TEP, SMS, Email, SoMed, ConMed, Tradit; Manual | RP Sim, RP Obs, HistY, Local | En, Fr, Pt, Ar |
| g_Attractve | Most attractive in West Africa: many desired features, similar to _h_Equipp_, but simpler distribution | HydObs: in situ WL, Q, EO WL; MetF: HydroGFD-WA, Am, Ens | Niger HYPE, WWH | Q, WL, P, E, SM, WQ | Excel, maps, graphs | Web, H-TEP, SMS, Email, SoMed, ConMed, Tradit; Manual | RP Sim, RP Obs, HistY, Local | En, Fr, Pt, Ar |
| h_Equipp | Fully equipped: all system elements, except recalibrated HYPE models | HydObs: in situ WL, Q, EO WL; MetF: HydroGFD-WA, Am, Ens | Niger HYPE, WWH | Q, WL, P, E, SM, WQ | Excel, maps, graphs | Web, H-TEP, FTP, API, SMS, Email, SoMed, ConMed, Tradit; choice (Automatic or Manual) | RP Sim, RP Obs, HistY, Local | En, Fr, Pt, Ar |
| i_Calibr | Recalibrated HYPE models | HydObs: none; MetF: HydroGFD2 | Recalibrated WWH | Q, WL, P, E, SM | Excel, maps, graphs | Web, H-TEP, SMS, Email, SoMed, ConMed, Tradit; Manual | RP Sim | En, Fr, Pt |
| j_Cal-EO | Recalibrated HYPE models and EO data | HydObs: EO WL; MetF: HydroGFD2 | Recalibrated WWH | Q, WL, P, E, SM | Excel, maps, graphs | Web, H-TEP, SMS, Email, SoMed, ConMed, Tradit; Manual | RP Sim | En, Fr, Pt |
| k_Cal- | Recalibrated HYPE models and EO data and in situ data | HydObs: in situ WL, Q, EO WL; | Recalibrated WWH | Q, WL, P, E, SM | Excel, maps, graphs | Web, H-TEP, SMS, Email, SoMed, ConMed, Tradit; Manual | RP Sim | En, Fr, Pt |





| EO- | MetF: Hy- |
|---|---|
| situ | droGFD2 |

### 3.4 Predicted performance of each system option

According to the expert predictions but excluding stakeholder preferences, none of the 11 FANFAR system options were able
to achieve the best level of all objectives (Figure 3; details see sect. SI-2.4, prediction matrix with raw data as input for MCDA
modelling see Table SI-30). Rather than indicating our failing to design the "perfect" system, it illustrates the impossibility to

design a perfect system, considering inherent tradeoffs between achieving objectives. For instance, the status quo pre-opera-
tional system option *a_Fast-dev* achieved the highest values for objective *31_short development time*, and *32_costs*, but scored
low on many other objectives such as accurate, clear, reliable, and timely information. Options that achieved high levels for
objectives of *High information accuracy and clarity*, inevitably need longer *development time* and have *higher costs*. There-
fore, it is not possible to clearly determine, which option is "best" based on only the predicted performance (Figure 3). We

require the input from stakeholders concerning which objectives are most important to them (sect. 3.5, sect. 3.6).

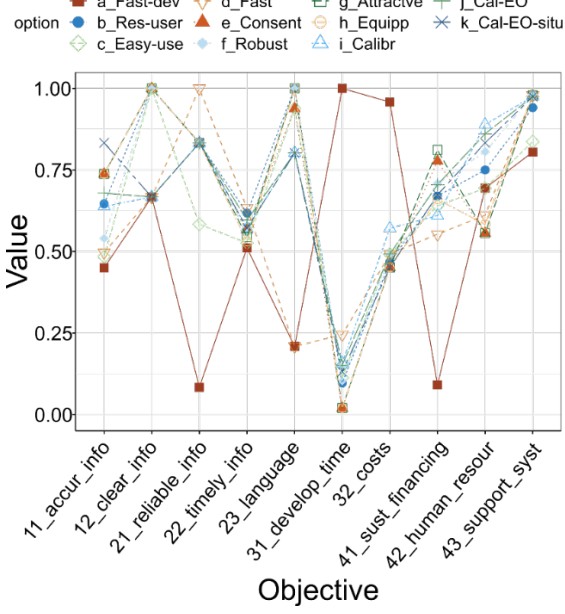

**Figure 3. Predicted value (y-axis) of 11 FANFAR system options (a–k; symbols) for 10 objectives (x-axis), based on expert predic-
tions, but not including stakeholder preferences. A value of 1 means that the option achieved the best level of this objective, and 0
that it achieved the worst level, given the ranges of the underlying attributes (i.e., it is a relative scaling from best to worst).**

### 3.5 Stakeholder preferences

The elicited weights (w) for the four higher level objectives were similar for all groups (w = total bar length; Figure 4), except
those of the French speaking emergency managers (G1A). These gave a high weight (w = 0.25) to *3. Low costs*, which was





least important for the others (0.1–0.12). G1A reasoned that all four higher level objectives are equally important in emergency

situations with a connected chain of events. In contrast, the higher level objectives *1. High information accuracy and clarity*,

and *2. Good information access* were generally regarded as most important by the other groups. There were some notable

differences in importance of lower level objectives. Again, the French speaking emergency managers (G1A) were exceptional

in assigning much lower weights to objectives they considered unimportant (objectives 23, 31, 41, and 43). They argued that

the goal in emergencies is to save lives, and FANFAR system development should focus on achieving fast access to flood

alerts (*22_timely_info*; 0.21), and on personnel that can deal with this information (*42_human_resour*; 0.25). The weight sets

in the other groups were overall more balanced (further details see sect. SI-2.6). There were different levels of consensus

within a group regarding the assigned weights, reflected in the length of the error bars (Figure 4).

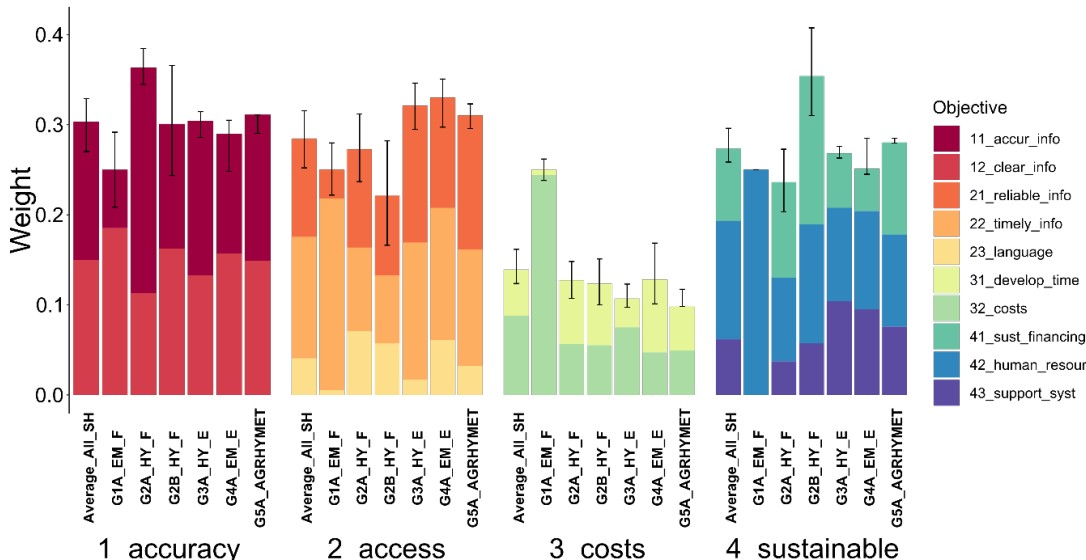

**Figure 4. Weights (y-axis) assigned to higher level objectives (blocks, *1_accuracy*, *2_access*, etc.) colored by weights of lower level objectives (*11_accur_info*, *12_clear_info*, etc.), averaged over all six stakeholder groups (Average_All_SH), and for each group**
**(G1A_EM_F, G2A_HY_F, etc; x-axis). Error bars: uncertainty of elicited preference statements, i.e., the sum of uncertainties of all lower level objectives within the branch of the respective higher level objective. Per definition all weights of a group sum up to 1.**

## 3.6 MCDA results

No FANFAR system option clearly outperformed the others for all stakeholder groups in the standard MCDA (setting S0;

Table 2) that did not consider uncertainty (Figure 5; details see Table SI-32; Table SI-33). The early stage option at the begin-

ning of the project (*a_Fast-dev*), achieved lowest total values (v < 0.46) and the last rank for all stakeholder groups except the

French speaking emergency managers (group G1A, v = 0.64, rank 5). This can be attributed to the different weight preferences

of G1A. All other options generally reached a high value for all groups, with only small differences between groups regarding

the performance of options. The total value ranged from v = 0.55 in the worst case (*d_Fast* for G2A) to 0.70 (*b_Res-user*,

G3A). Indeed, this option *b_Res-user* seemed somewhat better than the others, achieving a high value (v = 0.65–0.70) for all





groups, thus reaching the first rank for all, again with exception of group G1A, for which it still achieved the second rank. In
       other words, *b_Res-user* was able to achieve 65–70 % of the ideal case over all objectives in all stakeholder groups (for better
       understanding [0,1] values can be interpreted as percentages). Options *f_Robust*, *i_Calibr*, *j_Cal-EO*, and *k_Cal-EO-situ* also
       performed well (0.63–0.70) for all groups, while *c_Easy-use*, and *d_Fast* achieved the lowest values (0.55–0.64).

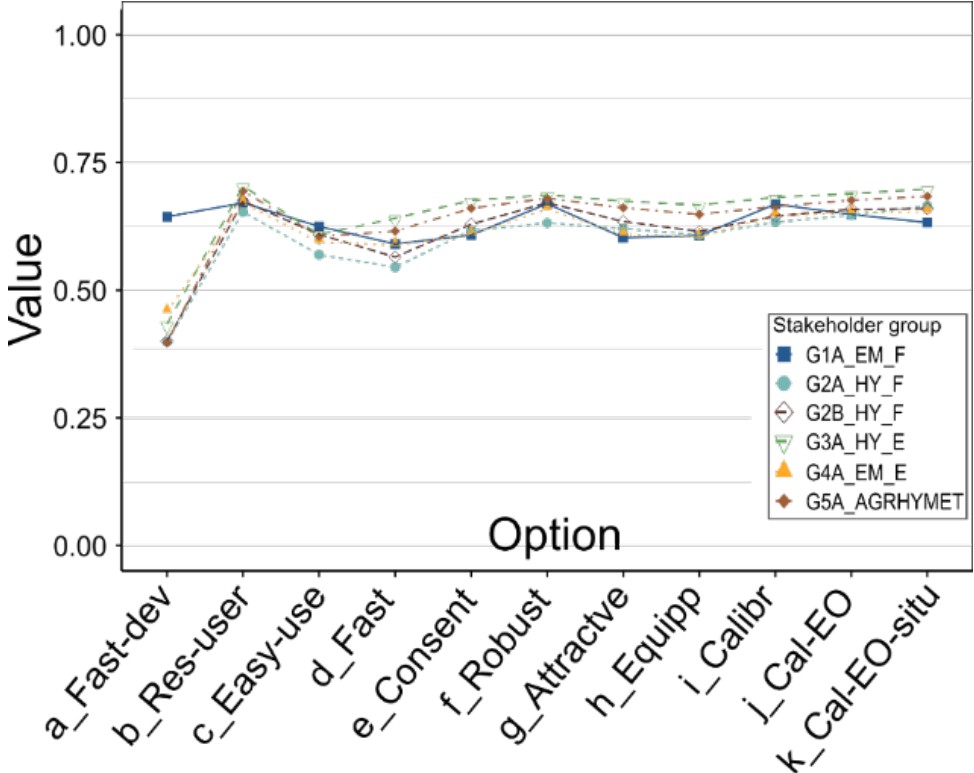

**Figure 5. Total aggregated value (y-axis) of 11 FANFAR system options (x-axis) for six stakeholder groups (symbols), without un-
       certainty. The higher the value, the better the option achieves the objectives, given expert predictions and stakeholders' preferences.**

       Including uncertainty of the expert predictions in the MCDA with Monte Carlo simulation clarified results. The FANFAR
       system options *b_Res-user* and *f_Robust* performed well and achieved the highest ranks for all stakeholder groups in the 1'000
       simulation runs (Figure 6; details see Table SI-34). The options *i_Calibr*, and *j_Cal-EO*, achieved good to medium ranks for

most groups in most simulations. Poor performance was achieved by options *a_Fast-dev* (again except group G1A), and
       *d_Fast*, which hit the last ranks in most simulation runs. The remaining options performed somewhere in between.

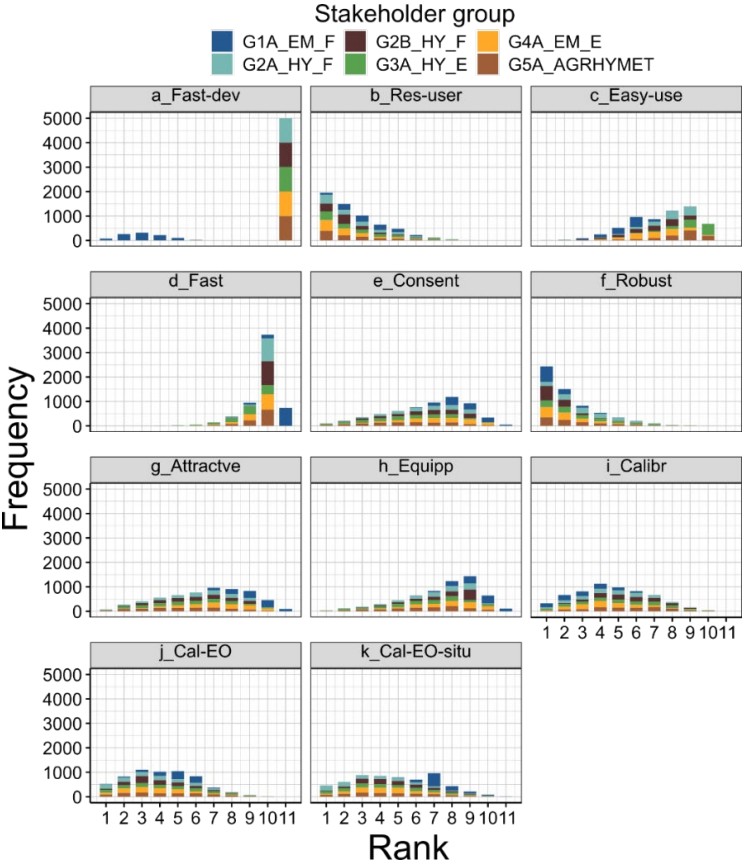

**Figure 6. Ranks of 11 FANFAR system options including uncertainty of expert predictions. Frequency (y-axis): how often each option (blocks, *a_Fast-dev*, *b_Res-user*, etc.) achieves a rank (1: best rank, 11: last rank; x-axis) in each model run for each stakeholder group (stacked bars). 1'000 Monte Carlo simulation runs drawing from uncertainty distributions of attribute predictions.**

### 3.7    Sensitivity analyses of stakeholder preferences

The performance of FANFAR system options was not sensitive to most changes in model parameters (Table 2). The least changes in rankings occurred between the standard MCDA (S0) and the sensitivity analyses that tested the extremes of the weight ranges elicited from stakeholders in the workshop sessions (S22–S232; Table 2). Here, Kendall's τ rank correlations were high, ranging from 0.86 to 1 (identical ranking of all options). Doubling the weight of *23_language* (S31) hardly impacted the options' rankings for any stakeholder group. Greater changes occurred using other aggregation models. The difference between our standard MCDA (S0) and changing aggregation models increased, the more the aggregation parameter γ increased from 0 (geometric mean; S12), over mixture models (S13, S14), to the additive model with γ = 1 (S11). Rank correlations were still relatively high between the additive model and our S0 standard model (0.53–0.86), and importantly, the rankings of the best-performing options, *b_Res-user*, and *f_Robust* did not change (details sect. SI-2.8). For other options, including *i_Calibr*, there were some greater differences, depending on the group. The greatest changes in options' rankings occurred for an alternative weight set (S21) elicited in the group of French speaking emergency managers (G1A). Interestingly, this alternative





set moved the rankings and values of system options to those of all other groups. Thus, based on this sub group, group G1A

was no longer an outlier, and e.g., option *a_Fast-dev* clearly performed worst for all groups, also G1A (Figure SI-40).

**Table 2. Results of local sensitivity analyses. Setting S0: default with elicited preferences of stakeholder groups and weighted power mean model, Eq. (3). Setting S11–S14: effect of other aggregation models (varying γ). S21–S22: uncertainty of Swing weights. S231–S232: uncertainty of Simos' card method weights. S31: increase (possibly underestimated) weight. S11–S31: all other parameters as S0. Columns G1A to G58: Kendall's τ rank correlation coefficient between ranks of options in main MCDA (setting S0) and ranks resulting from MCDA using other settings (S11–S31) for stakeholder groups (e.g., group G1A). Column mean: correlation between**
**S0 and average rank over all groups for which analysis was done. Note: S21 was only done for group G1A_E (i.e., mean = group correlation). Kendall's τ 1: identical ranks; 0: no correlation; –1: inverse relationship; –: not applicable. Kendall's τ from 0.81–1.00: underlined, indicating very good agreement between changed setting and S0; τ from 0.61–8.80 are dotted underlined.**

| Setting | Definition | G1A | G2A | G2B | G3A | G4A | G5A | Mean |
|---------|------------|-----|-----|-----|-----|-----|-----|------|
| S0 | Default. MCDA for all six stakeholder groups; $\gamma = 0.2$; see Methods, eq. (3) | | | | | | | |
| S11 | Additive model all groups; $\gamma = 1$ | 0.86 | 0.64 | 0.60 | 0.64 | 0.53 | 0.75 | **0.67** |
| S12 | Weighted geometric mean all groups; $\gamma \rightarrow 0$ | 0.96 | 0.78 | 0.93 | 1.00 | 0.82 | 0.93 | **0.90** |
| S13 | Mixture model; $\gamma = 0.5$ | 0.93 | 0.78 | 0.67 | 0.75 | 0.75 | 0.75 | **0.77** |
| S14 | Weighted power mean; $\gamma = 0.8$ | 0.89 | 0.67 | 0.64 | 0.64 | 0.53 | 0.75 | **0.69** |
| S21 | Alternative weight set for group G1A | 0.31 | – | – | – | – | – | **0.31** |
| S22_11_min | Weight ranges with $\Delta > 0.02$ from average weight for *11_accur_info*; minimum weight | 0.96 | – | 0.96 | – | – | – | **0.96** |
| S22_11_max | *11_accur_info*; maximum weight | – | – | 1.00 | – | – | – | **1.00** |
| S22_12_min | Weight ranges with $\Delta > 0.02$ from average weight for *12_clear_info*; minimum weight | – | – | 1.00 | – | – | – | **1.00** |
| S22_12_max | *12_clear_info*; maximum weight | 0.86 | – | 0.86 | – | – | – | **0.86** |
| S231 | Alternative weights resulting from ranges assigned to Z min | – | – | – | 0.89 | 0.89 | 0.89 | **0.89** |
| S232 | Alternative weights resulting from ranges assigned to Z max | – | – | – | 0.96 | – | – | **0.96** |
| S31 | Double weight of "Several languages | 0.96 | 1.00 | 0.93 | 0.93 | 0.89 | 1.00 | **0.95** |





Cost-benefit visualizations confirmed that options *b_Res-user*, *f_Robust*, and *i_Calibr* are suitable consensus options. For
summary visualization (Figure 7), we chose group G2A with typical weight preferences, and the outlier G1A (details sect. SI-
2.9). Again, group G1A had different results due to their different preferences. These three options are located on the efficient
frontier for all groups (two minor exceptions). Options below the efficient frontier should in principle not be considered be-
cause they are outperformed, given the visualization choice and the MCDA model including stakeholder preferences. For
instance, *d_Fast* was similarly expensive as *c_Easy-use*, *j_Cal-EO*, and *f_Robust*, which all achieved higher values for the
other nine objectives (Figure 7a). However, when including the objective *31_develop_time* in the costs (x-axis), the perfor-
mance of *d_Fast* improved (y-axis) for those groups who gave a higher weight to short development time such as G2A (Figure
7b). Some options such as *g_Attractve* performed well for most stakeholder groups, represented by G2A (Figure 7a; right), but
not for others (G1A, left). Obviously, the cheapest option (far right, x-axis), *a_Fast-dev*, the status quo option at project start,
also achieved the lowest value for the other nine (Figure 7a) or eight objectives (Figure 7b).

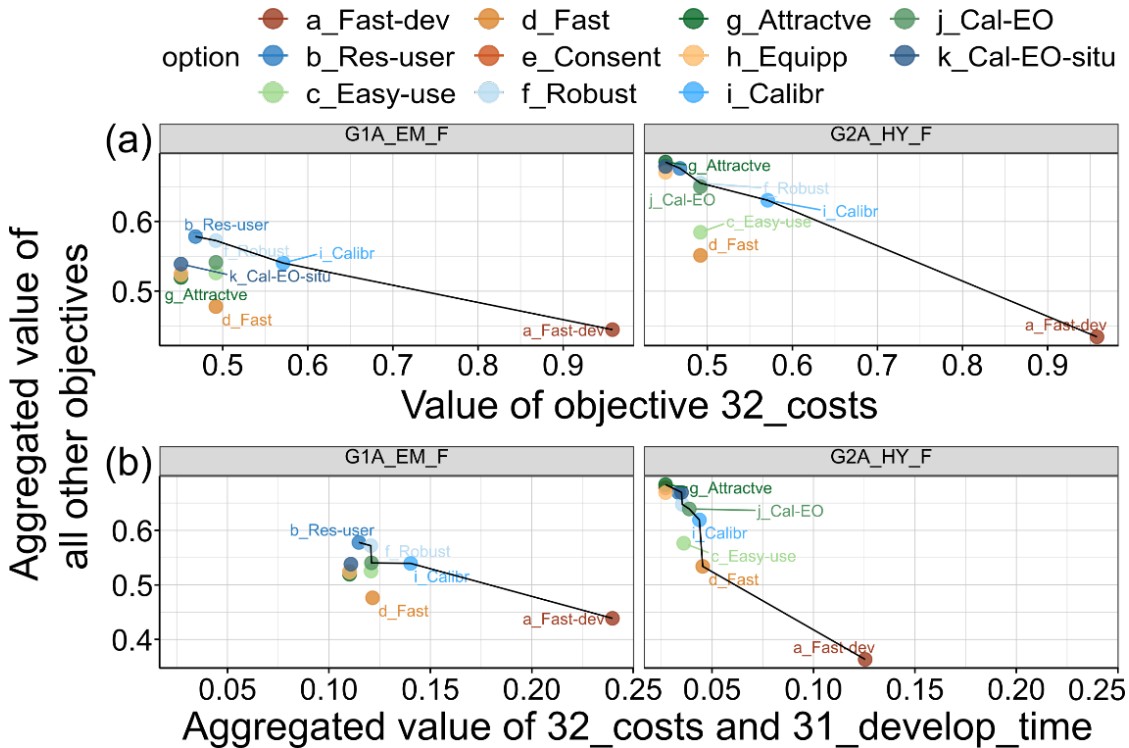


**Figure 7. Cost-benefit visualization of 11 FANFAR system options (colored dots) for stakeholder groups G1A_EM_F (left), and
G2A_HY_F (right). The costs (x-axis) are once plotted as the value of the total operation and maintenance costs (*32_costs*; top panel
a), and once as aggregated value of *32_costs* and the time needed to develop the FANFAR system (*31_develop_time*; panel b). Costs
are plotted against the total aggregated value of all other nine (panel a) or eight objectives (panel b, y-axis). Values range from 0**
**(worst case) to 1 (best), the highest cost options are to the left, and lower cost options to the right. Solid black line: efficient frontier
given the underlying MCDA model and chosen visualization, which exemplifies the best performing options for a given cost level.**





### 3.8 Stakeholders' perceived satisfaction with the FANFAR system

The majority of respondents perceived the current performance as sufficient for all objectives, based on both the direct question concerning use of the current system (b), and the inferred difference (c – a) between how much the current FANFAR system fulfills the respective objective (a) and the minimum acceptable fulfillment level (c). Across all objectives, 79 responses were positive, 16 negative, and 25 did not respond to question b. For the objective generally considered most important by stakeholders, *11_accur_info*, all respondents would use the current FANFAR system in future (Figure 8). However, four (out of 12) respondents indicated that the system does not currently meet their minimum acceptable performance requirements. This result is representative of the results for all objectives (details see sect. SI-2.10).

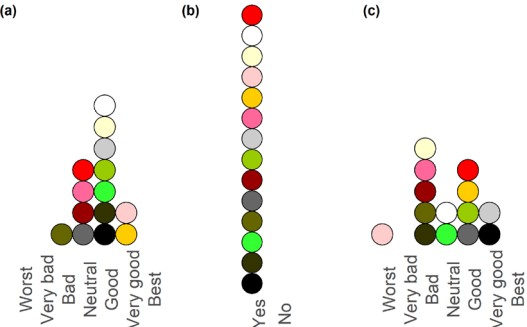

**Figure 8. Stakeholder perceived satisfaction with the performance of the FANFAR system during the 2020 rainy season for objective *11_accur_info*. Questions: (a) How much does the FANFAR system currently fulfil this objective? (b) Would you use the system in the future if it remains as is? (c) What is the minimum acceptable to you? Colored dots represent unique respondents (N = 12).**

## 4 Discussion

As the most important content related MCDA result, we were able to identify three FANFAR system options that achieved a good overall performance (Figure 5), despite uncertainty of expert estimates, model uncertainty, the existence of tradeoffs between objectives, and different preferences of stakeholders groups concerning the importance of objectives. One option that performed well, *b_Res-user*, was created by stakeholders in the first FANFAR co-design workshop. They chose system components with a view that *b_Res-user* requires the least resources for West African users in terms of skilled personnel, good internet connection, and stable power supply (Table 1). Option *f_Robust* was also created by stakeholders, with a similar perspective of reliably working under difficult West African conditions related to collecting in situ data and being able to distribute information through a wide range of channels. The third option *i_Calibr* is the most modest option created by FANFAR consortium members to capture interesting technical features using only calibrated models. Main improvements of *i_Calibr* are using refined HYPE models (e.g., with adjusted delineation and parameter calibration; Andersson et al., 2020b), but not including earth observation or in situ data (included in option *j* and *k;* Table 1). All three best options achieved 63–70 % of the ideal case across all objectives in all stakeholder groups. We consider this a very good value, given the tradeoffs that inevitably exist between fulfilling the different objectives. These options also emerged as sufficiently robust consensus options



from our extensive stress tests, namely (i) including the uncertainty of expert predictions with Monte Carlo simulation (Figure 6); probing the uncertainty of model assumptions and the stakeholders' weight preferences with sensitivity analyses (Table 2); and (iii) checking for dominance in cost-benefit visualizations (Figure 7). Interestingly, all three options do not incorporate the more advanced system features. This indicates that a flood forecast and alert system that meets the preferences of the West African stakeholders primarily needs to work accurately and reliably under difficult conditions in West Africa.

As the main practical result, the transdisciplinary MCDA framework allowed including stakeholders in structured, diverse, intensive, and sometimes also entertaining interactions in the co-design workshops. Their contributions and the close collaboration over several years shaped the development and important features of the FANFAR system. Research postulates that ownership, accountability, and legitimacy of solutions increases in collaborative transdisciplinary projects (Lang et al., 2012 and references therein). Because the stakeholders co-designed the FANFAR system, it should meet the main requirements of West African users, and hopefully, acceptance and future uptake in West Africa will be high. We discuss important phases in the transdisciplinary MCDA co-design process below (sect. 4.1), followed by some thoughts on lessons learnt (sect. 4.2).

## 4.1 Implications of results for the FANFAR project

### 4.1.1 Problem structuring phases of MCDA

Carrying out a relatively simple stakeholder analysis with questionnaires is effective for gathering important information. This includes finding out who has influence or power, and who is affected by (in our case) a good or malfunctioning flood forecast and alert system (Grimble and Wellard, 1997). As main insight, the important producers of flood information participated in the FANFAR co-design process, including hydrology representatives from every West African country, and key supranational organizations such as AGRHYMET (Table SI-4; details see Silva Pinto and Lienert, 2018). Mandated national emergency managers also participated in our workshops, and are main receivers of flood information. Thanks to their experience, we were able to integrate the alert dissemination chain in the FANFAR co-design process. Whether the current distribution channels such as the media are always effective in reaching intended recipients is a different issue, which we started exploring in sessions with emergency managers, including propositions for more effective warnings (Kuller et al., subm.). The stakeholder analysis also helped identifying missing parties. These include agriculture, industry such as dam managers and electricity utilities, or humanitarian aid organizations. We did not invite these because interactive workshops with more than 50 participants are too ineffective and challenging to manage. Nevertheless, some were attracted, and e.g., dam managers participated in informal dialogues at some workshops. Stakeholder analysis has been criticized for being done ad hoc, or for collecting laundry lists of concerns (Hermans and Thissen, 2009;Reed et al., 2009). A more rigorous approach is social network analysis (SNA), allowing to understand relationships between actors (Kenis and Schneider, 1991). There are benefits of combining qualitative data from stakeholder analysis with quantitative SNA (Lienert et al., 2013). For similar cases as FANFAR, we regard the simpler approach as sufficient if the main objective is to identify key organizations, and we recommend others to use it in their case.





Generating objectives in small groups with different methods ensured that a broad set of objectives was produced, thus avoid-
ing the common "group think bias" (Janis, 1972). We are confident that we captured the most important 10 objectives, which
cover the fundamental aims of the West African stakeholders (Figure 2). Recent research indicates that many environmental
applications of MCDA use too many objectives, the number ranging from three to 51 (Marttunen et al., 2018). This results in
a high proportion of objectives receiving a very low weight (lower than 0.05), which burdens later MCDA weight elicitation.
Thanks to consensus building discussions in plenary sessions where some objectives were excluded, we were able to avoid
this. Indeed, none of the objectives received an extremely low weight from any group (Figure 4; Table SI-31).

To generate options, solid technical knowledge of system components is needed. We could not assume that all participants
had this knowledge, but we aimed to avoid the "myopic problem representation bias", the tendency to stick to what one knows,
a typical issue in decision analysis (Montibeller and von Winterfeldt, 2015). We therefore used approaches that pre-structure
the process, but still allow for creative inputs of stakeholders' to foster thinking out of the box. We find the "Strategy Gener-
ation Table" (Gregory et al., 2012b;Howard, 1988) especially promising. In hindsight, knowing that many options performed
similarly well in the MCDA (Figure 5), it may seem as if some of the 11 created FANFAR system options (Table 1; Table SI-
6; Table SI-7) were too similar, and it might have been wise to search for a wider variety. However, the predictions, i.e.,
excluding stakeholder preferences, indicate that there indeed exist some substantial differences between the performances of
options for certain objectives (Figure 3). For instance, there was a relatively good spread concerning the achievement of the
two objectives of *1. High information accuracy and clarity* (*11_accur_info*; *12_clear_info*), among others. This indicates that
the MCDA aggregation step over different objectives including stakeholder preferences is responsible for the resulting simi-
larity between overall performances of most options. Moreover, we wish to point to a misunderstanding between some option
names and the practical implementation of selected system components. We decided to keep the original names. For instance,
*b_Res-user* is described as requiring the "least resources for users", while including "in situ gauge observations" would require
substantial resources from national or regional hydrological agencies to operate, collect, systematically store, and make new
observations available to the forecasting system every day. This does not impact the MCDA result; the option names are
irrelevant in MCDA preference elicitation, which focuses exclusively on the objectives. The expert predictions also focus on
how well each objective is achieved by each option, which is shaped by the specific components, irrespective of the name.
Multi-Attribute Value Theory (MAVT; Eisenführ et al., 2010;Keeney and Raiffa, 1976) allows including new options in later
MCDA phases (Reichert et al., 2015). We took advantage of this by creating additional options that cover certain technical
aspects considered potentially important by hydrologists of the FANFAR consortium, including e.g., ensemble meteorological
forecasts, redelineation and calibration of hydrological models, assimilation of EO water levels, and assimilation of in situ
gauge observations from rivers across the continent (options *h* to *k*, Table 1; Table SI-6). The system option describing the
state at the project start *a_Fast-dev*, was also created during post-processing. It captures the status quo, and can serve as
benchmark for monitoring successful system development. Indeed, it performed poorly for most groups (Figure 5).



We emphasize that the FANFAR system was continuously improved during the co-design process, also after eliciting stakeholder preferences in the second workshop. It now combines different aspects of the MCDA options. Currently (March 2021), the system is closest to *b_Res-user* (Table 1) regarding data inputs (assimilating in situ gauge observations, relying on the improved global meteorological forcing data HydroGFD3), distribution channels (interactive web visualization, SMS and Email alerts), support (knowledge base, help desk, forum, and a group on social media), and data download (table for selected stations). The current system also includes key elements of *i_Calibr* (deploying several refined HYPE models, providing model performance information). However, some elements are only partly implemented, e.g., the language of the web visualization interface (English, French) and the core ICT platform (English) not yet being available in Portuguese and Arabic. Some aspects still remain as in option *a_Fast-dev* such as the forecast variables (streamflow only), no public access to observations, distribution automatization (automatic), and the reference flood hazard threshold (return periods based on simulations). Reasons are partly legal (e.g., right to publish in situ observations), but mostly related to constraints in development time (e.g., multiple variables and alternative thresholds were conceptualized and programmed, but not yet operationalized).

Our stakeholder interactions based on the MCDA framework were motivated by "Value focused thinking" (Keeney, 1996). We emphasized the objectives, rather than starting with FANFAR system options. This proved beneficial, because asking: "What is important when designing a flood forecast and alert system for West Africa?" already in the first co-design workshop allowed the system developers to focus on priority objectives, irrespective of the later MCDA results. Thereafter, elicitation of stakeholder preferences in the second workshop guided system design. To give one example: in earlier discussions, some stakeholders voiced the strong opinion that the FANFAR system needs to be available in several languages, preferably also in local African languages. However, when asked to make tradeoffs between different objectives during weight elicitation, it became clear that achieving other objectives should be prioritized. Moreover, a flood forecast and alert system is technically complex, composed of many elements (Andersson et al., 2020b;Arheimer et al., 2011;Emerton et al., 2016). Focusing on the system per se, there is a risk of getting lost in stakeholder discussions about system elements. Instead, we included any stakeholder suggestion when creating options without forcing a choice. It was then up to the hydrologists and ICT specialists to find best solutions, given the stakeholders' priorities and the experts' predictions about system performance.

### 4.1.2    Dealing with uncertainty of predictions, preferences, and model assumptions

Attributes operationalize achieving objectives (Eisenführ et al., 2010;Keeney and Raiffa, 1976). Seemingly trivial, this is often challenging. We illustrated this for *11_kge*, constructed from the KGE index for 1, 3, and 10 day forecasts to measure the achievement of *1.1 High accuracy of information* (sect. 2.6). The data input uncertainty of the expert predictions was relatively large for some attributes (e.g., *11_kge*, *22_time*, or *42_experts*), but small to inexistent for others (*12_info*, *23_langue*; see boxplots Figure SI-30). The resulting overall uncertainty affected the results less than might be expected (Figure SI-35).





The weights indicate that most groups preferred that the system produces accurate, clear, and reliable information that reaches recipients well before a flood (*11_accur_info*; *12_clear_info*; *21_reliable_info*; *22_timely info*; Figure 4). Furthermore, capability for West African countries to interpret and handle the information was considered important by all groups (*42_human_resour*). We captured differences within groups about exact numbers with uncertainty ranges or a separate preference set

within groups (e.g., subgroups G1A, G2B; see sect. SI-1.2.3; sect. SI-2.6). Most importantly, the French speaking emergency managers (G1A) had strikingly different preferences compared to all other groups. Their weights were the result of intensive discussions during weight elicitation (this goes for all groups). Interestingly, all groups considered several languages as unimportant in weight elicitation, despite emphasizing in general discussions that language diversity is crucial. When asked to make tradeoffs between the possibility of having an inaccurate flood forecast and alert system, they were willing to renounce from

being able to choose the language. Similarly, they were willing to tradeoff higher operation and maintenance costs (except group G1A) and development time in return for receiving a well-functioning and precise system.

Including the uncertainty of expert estimates and stakeholder preferences in MCDA can blur results, and more elaborate models have been proposed (e.g., Haag et al., 2019b;Scholten et al., 2015). For FANFAR, simpler Multi-Attribute Value Theory and local sensitivity analyses (e.g., as in Zheng et al., 2016) were sufficient, since they enabled identifying options suiting all

stakeholder groups. Interestingly, the frequency rank plots indicate that including the uncertainty of predictions helped to *better* distinguish between performances of options (Figure 6), compared to the standard analysis without uncertainty (Figure 5). Options *b_Res-user* and *f_Robust* consistently achieved the first ranks in 1'000 simulation runs, and e.g., *i_Calibr* good to medium ranks. Some favorite options such as *k_Cal-EO-situ*, also ranked last in numerous runs (Figure 6), despite achieving good values when uncertainty was disregarded (0.63–0.70; Table SI-33). It was also evident that *a_Fast-* and *d_Fast_dev*

would be an imprudent choice, because they ranked last in most runs (except option *a* for the outlier group G1A).

Local sensitivity analyses confirmed that options *b_Res-user*, *f_Robust*, and also *i_Calibr* are robust choices. Changing stakeholder preferences hardly changed the MCDA results compared to our standard model (S0; Table 2). Doubling the weight of *23_language* (S31) did not affect results in all groups, which is good news for FANFAR system developers, since translation of all system components to several languages entails high costs. Operation and maintenance costs would have been another

candidate for doubling the weight, but was already covered by the high weight given by group G1A_EM_F. For this group with different preferences, the sensitivity analyses on the weight ranges given by group participants with a different opinion (S21; Table 2) changed the results in such a way that they aligned with the results of the other stakeholder groups, which is again good news. Moreover, we challenged our MCDA model assumptions and found that the additive aggregation model (Eq. (1); sect. 2.8), did indeed impact the rankings of the FANFAR system options (Table 2). As standard model, we assumed non-

additive aggregation (Eq. 3), relatively close to a weighted geometric mean model, based on feedback in the weight elicitation sessions. After discussing some examples, all groups stated that poor performance on an important objective should not be compensated by good performance on others, a main implication of additive aggregation. For decision analysis, this is yet another recent case confirming that the additive model can unintentionally violate stakeholder's preferences (e.g., Haag et al.,





2019a;Reichert et al., 2019;Zheng et al., 2016). Consequently, using additive aggregation as default may not be the best choice.

It is the uncontested model in most MCDA applications because it is easy to understand, elicit preferences, and implement. We have overcome the latter barrier by allowing for fast change of model parameters in the ValueDecisions app (Haag et al., in prep.). However, in some cases it may be important that stakeholders understand the underlying model. In a recent paper about behavioral issues in modelling, Hämäläinen (2015; Table 2) recommends using "transparent and simplified models for learning (when modelling together with stakeholders), and comprehensive models for problems solving". For FANFAR, local

sensitivity analyses sufficed to conclude that additive aggregation has an effect, but does not alter the ranking of the best performing options. As final confirmation, we used cost-benefit visualization (e.g., Liu et al., 2019). Also here, options $b\_Res$-$user$, $f\_Robust$, and $i\_Calibr$ (Figure 7) were consistently on the efficient frontier, including the original weights of group G1A. This means that no other option performed as well as these on all remaining objectives (i.e., "benefit") for given *operation and maintenance costs*, but also when including *development time*, or *sustainability of long term financing* in the costs. From all

these analyses we can safely conclude that the proposed three options are good configurations of the FANFAR system.

### 4.1.3    Stakeholder feedback

The online survey allowed receiving some feedback about how well the current FANFAR system meets the 10 objectives, despite not being able to carry out a workshop in Africa due to the COVID-19 pandemic. The stakeholders were quite satisfied with its performance during the 2020 rainy season (Figure 8), and are generally willing to use the FANFAR system in future.

## 4.2    Challenges of transdisciplinary process and lessons learnt

### 4.2.1    Collaborative problem framing phase

To discuss our learning, we draw from the design principles of transdisciplinary research processes in Lang et al. (2012), and identified challenges (Table 3 in Lang et al., 2012; marked bold hereafter). In the first phase of transdisciplinary projects, highly relevant challenges for FANFAR include "***Unbalanced problem ownership***" and "***Insufficient legitimacy of actors***

***involved***". To avoid this, FANFAR integrated two key West African stakeholders in the consortium: AGRHYMET (supra-national institution mandated by 13 West African member states and ECOWAS to provide e.g., operational flood warnings), and NIHSA (Nigerian Hydrological Services Agency). AGRHYMET can serve as intermediary (boundary) organization between research and implementation in West Africa (Lang et al., 2012). Additionally, stakeholder analysis helped to identify the most important participants for the co-design process. Participation of representatives from 17 national (and other regional)

hydrology or emergency services should help avoid unbalanced problem ownership and insufficient legitimacy, since the FANFAR system is potentially relevant for their institutional core mandate. Of course, the problem remains that being fully inclusive in a large region like West Africa is unrealistic, given our identification of 68 distinct stakeholder types (sect. 3.1).





### 4.2.2    Producing new knowledge

Challenges of the second phase include *"Conflicting methodological standards"*. This may concern collaboration between
technical experts, hydrologists, and decision analysists oriented on social sciences, or European and African hydrologists.
Coping strategies include the *"Use of demonstration projects"* (Lang et al., 2012), such as the FANFAR system. We fostered
extensive experimentation with the current stage of the pre-operational system in each workshop. We elicited structured feed-
back for co-designing system elements from many different experts. Moreover, we tried to overcome *"Lack of e.g., technical
integration"* by using MCDA, which allows combining qualitative and quantitative data (i.e., integrating predictions and pref-
erences). MCDA thus supports better understanding between social, technical, and natural science disciplines, a pertinent issue
in any interdisciplinary project (Gregory et al., 2012b). To give one example, in an ecosystem services project, scientists sought
quantitative data, whereas for policy and decision making, also qualitative aspects were important (Lang et al., 2012).

Definitely a challenge in FANFAR was *"Discontinuous participation"*. The project runs over three and a half years, involving
representatives of at least two institutions from 17 West and Central African countries, regional stakeholders, and European
partners. This is reflected in the changing numbers of participants in the co-design workshops (sect. 2.2), and unfortunately in
changing composition of participants. Possible coping strategies are *"Low thresholds for and appropriate levels of participa-
tion"* (Lang et al., 2012). However, as an example given by these authors, we encountered the opposite challenge: requests for
participation increased over time, and we had to keep numbers at a manageable level. We attempted to integrate new partici-
pants by presenting the current state of the FANFAR system, and e.g., the MCDA objectives, at each workshop. We also
invited all workshop participants to present the use of the FANFAR system in their country at every workshop, irrespective of
participating before. For MCDA, discontinuous participation was not overly problematic, as new participants in the second
workshop accepted the objectives (Figure 2) and options (Table 1). This indicates that our sample of participants was probably
sufficiently large to be able to identify appropriate objectives and options among the participating stakeholder types. Preference
elicitation was carried out only during the second workshop, and the weights (Figure 4) are thus based on those preferences.

We sought to overcome *"Vagueness and ambiguity of results"* by using MCDA, which can provide clear results, even in face
of uncertain data (sect. 4.1.2). The recommended coping strategy *"Specification and explicit conflict reconciliation"* (Lang et
al., 2012), forms an integral part of MCDA. Opposing interests of stakeholders do not need to end up in conflict about the
choice of options, but are valued as part of the methodology (examples see Arvai et al., 2001;Gregory et al., 2012a;Gregory et
al., 2001;Marttunen and Hamalainen, 2008). In the weight elicitation sessions, we encouraged stakeholders to discuss diverg-
ing preferences (sect. 2.7.2), and we recommend allowing for this type of uncertainty. First, it helps participants to construct
their own preferences, as these are usually not readily available in our mind (Lichtenstein and Slovic, 2006). In fact, we had
been concerned that weight elicitation might be cognitively too demanding. This seemed not to be the case, as there was no
difference to earlier experience from workshops in Europe. Some participants quickly grasped the method and helped others.
Thereafter, participants explained and defended their opinion in lively discussions until either consensus was found, or the
moderator recorded the diverging preferences for inclusion in later sensitivity analyses. Second, discussion without dispute



enables learning from peers and understanding alternative perspectives. Third, conflicting preferences can inform later sensitivity analyses (sect. 2.9.2). In our case, conflicting preferences did not change the ranking of options (sect. 4.1.2). In other cases, they can help constructing better options.

Regarding methods, we were interested in the opportunities and limitations posed by the West African setting. We used dif-
ferent methods for generating objectives (sect. 2.4) and FANFAR system options (sect. 2.5). Filling in pen and paper surveys individually proved more arduous for many participants than group interactions. While some finished the surveys fast, others needed much time and guidance. It remains unclear whether this was caused by questionnaire complexity, length, or clarity of our presentations, or unfamiliarity with surveys among people in West Africa, and requires attention in future studies. Online surveys were even more problematic due to internet and computer problems. We recommend considering this when working
in locations with similar issues. In contrast, all interactions in small groups were fruitful, with engaged and animated discussions (e.g., brainstorming, brainwriting-635, weight elicitation). For generating options, the strategy generation table, an interactive but very structured approach, and giving clear examples (e.g., exemplifying forecast variables as "streamflow" or "water level"), worked better than less structured approaches (e.g., asking about "forecast variables"). This is known from brainstorming objectives, which may be inferior to methods giving more guidance (Bond et al., 2008;Haag et al., 2019c). For plenary
discussions, we recommend good moderation and structured methods to collect inputs. For instance, the structured session where participants discussed and voted on objectives delivered clear results, while plenary sessions generally requesting feedback produced less information. We have similar experiences from earlier workshops in Switzerland, Portugal, and South America. One difference between the continents is, however, the perceived legitimacy of democratic voting. Some of the West African stakeholders did not view the individual voting system as fully legitimate, as it does not weigh votes according to e.g.,
existing hierarchies in and among stakeholder institutions.

### 4.2.3    (Re-)integrating and applying the co-produced knowledge

A major challenge experienced in many transdisciplinary projects is inadequate or missing generalization of the ***"Limited case specific solution options"*** (Lang et al., 2012). As a consequence, generated knowledge does not add to scientific progress, and is not adopted in societal praxis in similar projects. This paper together with other project outputs (FANFAR, 2021) hopefully
help overcome this problem. For the MCDA, we document the process, methods, results, and our interpretation and experiences, including extensive details in the Supplementary Information. We encourage hydrologists to use this material. We wish to emphasize that it is not necessary to carry out a full MCDA in every case. Problem structuring alone (discussed in sect. 4.1.1) can create many useful insights, and may be easier to apply.

A final challenge is ***"Tracking of scientific and societal impacts"***, as standardized approaches to evaluate the outcomes of
transdisciplinary research are still missing. This requires follow-up studies, which are integrated in current proposals for full operationalization of the FANFAR system. AGRHYMET, mandated by ECOWAS to provide flood forecast and alert information across West Africa, has the authority to drive the uptake and operationalization of the FANFAR system. The project





included several components to facilitate long term sustainability (e.g., co-development, open source tools, documentation, training material, and capacity development activities). It is encouraging that AGRHYMET already uses the FANFAR system
beyond project activities (e.g., at the PRESASS and PRESAGG forums (WMO, 2021), to support the ECOWAS flood management strategy, and in their MSc curriculum). Nevertheless, long term sustainability and operationalization after termination of EU sponsoring is still not secured. AGRHYMET now leads the effort to secure financing for the FANFAR system, along with other consortium partners. As this challenge is not listed in Lang et al. (2012), we propose adding it to the framework.

## 5    Conclusions

We documented what we believe to be an interesting and important example of a transdisciplinary, transcontinental co-design process in a complex setting: producing a good flood forecast and alert system for West Africa. The MCDA framework was useful to structure and guide our interactions with the West African stakeholders. MCDA integrates a variety of methods, both participatory methods in the problem structuring and preference elicitation phases, and more standard science methods for integrating data. In such a project, uncertainty is inevitable. We included uncertainty of expert predictions with Monte Carlo
simulation and uncertainty of model assumptions and stakeholder preferences with sensitivity analyses. The MCDA results indicate that uncertainty plays a role and that its inclusion can be beneficial. It allows integrating the values and preferences of stakeholders without forcing consensus. As additional and unexpected insight, uncertainty even helped identifying three robust FANFAR system options. These well performing options are the "tangible product" (Lang et al., 2012) of the transdisciplinary MCDA process. Interestingly, these best performing flood forecast and alert system options are not the most techni-
cally elaborate options that we analyzed. Stakeholders designed them to work well under challenging West African conditions, such as lack of up to date and accurate in situ observations, internet and power supply cuts, and lack of skilled personnel. Moreover, they meet stakeholder preferences. The transdisciplinary FANFAR project also produced intangible outcomes, such as learning and capacity building for decision making among participants. Finally, it is our strong belief that transdisciplinary research both contributes to solving the complex problems our world is facing, and to advance scientific progress. We therefore
encourage our colleagues conducting water research to engage in transdisciplinary projects with stakeholders and society.

## 6    Appendices

None

## 7    Data availability

The data will be available on the Eawag Research Data Institutional Collection (ERIC: https://opendata.eawag.ch)

785                    < DOI Link to be provided once paper is accepted >





## 8 Supplement link (< will be included by Copernicus, if applicable >)

In the Supplementary Information (SI), we provide ample material to guide readers unfamiliar with Multi-Criteria Decision Analysis (MCDA) through the decision making steps. This includes a Methods section (generating options, eliciting weights, MCDA model, sensitivity analyses, stakeholder feedback), and a Results section (stakeholder analysis, objectives and attributes, system options, predictions, marginal value functions, weights, MCDA results, and stakeholder feedback).

## 9 Author contribution

Judit Lienert: Conceptualization, funding acquisition, investigation, methodology, project administration, resources, supervision, validation, writing – original draft preparation. Jafet Andersson: Funding acquisition, investigation, project administration, writing – review & editing. Daniel Hofmann: Data curation, formal analysis, visualization, validation, writing – original draft preparation. Francisco Silva Pinto: Formal analysis, investigation, methodology, project administration, writing – review & editing. Martijn Kuller: Investigation, project administration, supervision, writing – review & editing.

## 10 Competing interests

The authors declare that they have no conflict of interest.

## 11 Special issue statement

This paper was prepared for the special issue: "Contributions of transdisciplinary approaches to hydrology and water resources management"

< The statement on a corresponding special issue will be included by Copernicus, if applicable >

## 12 Acknowledgements

This project has received funding from the European Union's Horizon 2020 research and innovation programme under grant agreement No. 780118: FANFAR (Reinforced cooperation to provide operational flood forecasting and alerts in West Africa (FANFAR, 2021). We thank Eawag, the Swiss Federal Institute of Aquatic Science and Technology, for providing additional funding to support the project during the COVID-19 prolongation period. We sincerely thank Abdou Ali, Addi Shuaib Olorunoje, Aishat Ibrahim, Alice Aubert, Aytor Naranjo, Berit Arheimer, Bernard Minoungou, Bode Gbobaniyi, Cletus Musa, David Gustafsson, Emilie Breviere, Emmanuel Mathot, Fabrizio Pacini, Fridolin Haag, Kevin Schönholzer, Léonard Santos, Lorna Little, Melissande Machefer, Mohamed Hamatan, Philipp Beutler, Sara Schmid, Tharcisse Ndayizigiye, and Umar Magashi for their invaluable support when preparing the workshops, collecting data during workshops, and data analysis. We are




truly grateful for the inspiring collaboration with the workshop participants, and we wish to sincerely thank them for their kindness and enthusiasm to work with us.

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
