# Peer review of "Using Multi-Criteria Decision Analysis for transdisciplinary co-design of the FANFAR flood forecasting and alert system in West Africa"

_Hydrology and Earth System Sciences, 2021_

## Author Comment (AC1)

Eawag    Environmental Social Sciences (ESS)
Überlandstrasse 133    Dr. Judit Lienert
P.O. Box 611    Cluster Leader Decision Analysis
8600 Dübendorf    +41 (0)58 765 55 74
Switzerland    judit.lienert@eawag.ch
+41 (0)58 765 55 44    https://www.eawag.ch/en/aboutus/portrait/organisation/staff/profile/judit-
www.eawag.ch    lienert/show/

[Figure]

Dübendorf, 23 June 2021

**Response to referee # 1**

Dear Referee

Thank you very much for reviewing our manuscript:

**Judit Lienert, Jafet Andersson, Daniel Hofmann, Francisco Silva Pinto, Martijn Kuller, "Using Multi-Criteria Decision Analysis for transdisciplinary co-design of the FANFAR flood forecasting and alert system in West Africa". hess-2021-177**

This manuscript was written for the HESS Special Issue **"Contributions of transdisciplinary approaches to hydrology and water resources management"**

We are grateful for the work that has gone into reviewing our paper. We do know that this takes a lot of time, which receives no direct reward. We are very willing to improve the manuscript based on your inputs, wherever possible.

We have addressed your comments one-by-one below. *The referees' comments are given in Italics*, our response is given in normal font.

We look forward to suggestions for improving the manuscript so that it meets requirements of publications in HESS.

With best regards,

Judit Lienert

also on behalf of my co-authors, Jafet Andersson, Daniel Hofmann, Francisco Silva Pinto, and Martijn Kuller

**Anonymous Referee #1**

*1) This manuscript does not meet the standards of a good research paper. It more like a diary or a story of what was done.*

**Response:** We are sorry that referee #1 does not think that our manuscript meets **scientific standards**. However, to us it is unclear what the referee means with "standards of a good research paper."

We do see that the manuscript can be read as "a diary or story of what was done". Our aim was to address the call of the Special Issue on Contributions of transdisciplinary approaches to hydrology and water resources management, for which the paper was written: "While interdisciplinary conversations have been happening to some extent, **transdisciplinary endeavours remain largely undocumented**. The type of transdisciplinarity we are interested in here engages, broadly speaking, academic and non-academic perspectives in knowledge production." (See the call of the SI)
* * *
*2) It is also quite annoying that the authors are so strong agenda in advocating the terms transdisciplinary and co-creation without really demonstrating what new those ideas bring into the traditional MCDA process.*

**Response:** We did not intend to bring **new ideas to the "traditional MCDA process"**. Rather, we aimed to verify in practice, whether a structured, participatory MCDA process, which is carried out in close interaction with a large number of African stakeholders at different steps of the MCDA, is a good/useful "approach for carrying out a transdisciplinary endeavor in hydrology and water resources management" – again, following the call of the SI.
* * *
*3) There are lots of unjustified claims and procedural statements. There is no clear structure of the overall modelling procedure and methods used. New twists and approaches are introduced here and there along the text. The structure used in the modelling seems to be very different from standard procedures so a description and justification is needed in one place not scattered in the text.*

**Response 3a):** We are sorry that referee #1 regards the **modelling procedure and methods as unclear in structure**. Please see Figure 1 in the paper (also see below), showing the MCDA structure. In our opinion, this is clear (and standard procedure). The sections of the paper (Methods, Results, Discussion, and Supplementary Information) follow this stepwise structure. In Figure 1, additionally to the MCDA steps, we cross-reference to the "transdisciplinary design" approach, in order to bridge between these two strands of literature.

[Figure]

**3b)** Referee #1 states that there are **many "unjustified claims and procedural statements"**, without specifying, which. He/she claims that **"new twists and approaches are introduced", which are very different from "standard procedures"**, without defining what he/she means with "standard procedures". We define our procedure in Section 1.3: Multi-Attribute Value Theory (MAVT), referencing classic textbooks (MAVT) (e.g., Eisenführ et al., 2010; Keeney, 1982; Keeney & Raiffa, 1976). In the methods section of each step, we provide more details on the exact «how», also citing scientific literature, e.g., in the «problem structuring steps 1 – 4 (Fig. 1). Some of our approaches are often overlooked in simpler MCDA approaches, including stakeholder analysis (section 2.3; see Lienert et al. (2013)), a well-founded generation of objectives (section 2.4; see e.g., Bond et al., 2008; Haag, Zuercher, et al., 2019), or the choice of an appropriate MCDA aggregation model that reflects the preferences of stakeholders (section 2.8; see e.g., Haag, Lienert, et al., 2019; Reichert et al., 2019). We shortly summarize this newer literature in the respective sections. Some more detailed method descriptions, which would have made the manuscript even longer, are presented in the Supplementary Information.

Of course, we can try to summarize «standard MCDA» and «novel MCDA» elements, but this strongly depends on the perspective of the reader, making a classification difficult. However, if regarded as suitable (e.g., by the editors), we could try to put this information in an overview table.
* * *
*4) The essential question is whom is the paper intended to and what are its real contributions? As it is now the modelling parts can possibly be understood by someone who is well familiar with different MCDA methodologies. I do not think that the readership of this journal has the required background in MCDA. The contribution can be that such an extensive project has been set up and completed. But for a project description there should also be some*

*critical evaluations of the possible weaknesses and challenges related to the modelling approaches used. Now the report only provides a happy story. This is never the case in real life.*

**Response 4a): Our aims** are presented in section 1.1: "(i) to exemplify the use of Multi-Criteria Decision Analysis (MCDA) as methodological framework for integrating stakeholders in a structured co-design process; (ii) to prioritize development of suitable FANFAR flood forecast and alert system configurations based on expert estimates about system performance as well as stakeholder preferences; and (iii) to document empirical evidence of a large transdisciplinary, transcontinental co-design process, and discuss insights, lessons learnt, and recommendations of special interest to hydrology praxis when engaging with stakeholders and society." We think that the aims match the call of the SI.

If a problem of the manuscript is that our research aims are not formulated as **research questions**, we propose to re-formulate the aims as specific research questions, and would **re-structure** the paper to answer these, e.g. along these lines:

(i)     **What would characterize a good regional-scale flood forecasting and alert system for West Africa?** In other words, how do the MCDA results contribute to finding a suitable FANFAR system configuration? Is it possible to find a "good compromise system configuration", despite large uncertainty (of expert predictions about FANFAR system performance, and of MCDA model), and despite the fact that stakeholders may have strongly different preferences about what the system should look like?

(ii)    **How can a large number of stakeholders be integrated** into a transdisciplinary process aimed at designing a flood forecast and alert system for entire West Africa (FANFAR system)?

(iii)   How much do the early problem structuring steps help **focusing the development of the FANFAR system at the beginning of the project to meet the stakeholders' expectations** (i.e., before the MCDA results are available)? How well does this early focus match later MCDA results?

(iv)    **What worked well, what worked less well?** What insights, lessons learnt, and recommendations can we provide to hydrology praxis when engaging with stakeholders and society?

We kindly ask for feedback (e.g., by the editors), whether this restructuring and reformulation into research questions would be helpful?

**4b) Intended audience.** The paper is intended to reach readers of HESS, not familiar with MCDA, as stated in section 1.3: "Additionally, we attempt to present the MCDA methods such that they are easily accessible and adaptable to other transdisciplinary projects, e.g., in hydrology. Extensive details are provided as blueprint in the Supplementary Information". We think that readers interested in carrying out an own MCDA receive the essential information regarding the different steps needed, and the literature that specifies each step in detail. We would not describe the MCDA process such detail in an Operational Research journal.

**4c) Contribution of the paper.** See point 4a) above. Additionally, we **contribute to documenting transdisciplinary research projects**. We are not aware of many transdisciplinary projects that carry out a thorough MCDA process with that many stakeholders, from 17 countries, in several large workshops (50 – 60 participants), in hydrology, in Africa (or in similar contexts). However, we have not reviewed this literature. We would be willing to do so, if the

editors or referees consider this type of review useful. We do wish to point out, that such additions will again add to the length of the manuscript.

**4d) Critical evaluations.** We extensively discuss challenges that we encountered in the Discussion section. However, we emphasize that the point in this paper is NOT primarily about the modelling approaches, but the challenges of interacting with stakeholders in order to co-design[1] a "good" FANFAR flood forecast and alert system for entire West Africa (based on a structured MCDA procedure, which includes modelling). We thus focus on the challenges of a transdisciplinary project – a large real-world project, whose outcome can potentially have large real impacts on people living in West Africa. The challenges thus include MCDA modelling as well as social science or transdisciplinary aspects of stakeholder participation:

- how to engage with a large number of stakeholders in an effective and productive way to produce a single flood forecast and alert system that works in entire West Africa (see Discussion section 4.1.1)
- how to create a common set of objectives that meet the diverse stakeholders values and expectations
- including the (possibly strongly) different preferences of stakeholders (in the model) with the aim of finding ONE FANFAR system configuration that meets these differing preferences (trade-offs between objectives are inevitable)
- how to deal with uncertainty (in the model): of expert predictions and stakeholder preferences, and model assumptions (we did this with Monte Carlo simulation and extensive sensitivity analyses; see Discussion section 4.1.2)
- how to deal with problems that have been identified as typical for transdisciplinary processes (Lang et al., 2012), including those in the collaborative problem framing phase, the production of new knowledge, and re-integration and application of the co-produced knowledge (see Fig. 1; Discussion section 4.2)
- and specifically, how well all of this works in the West African setting.

The **Discussion section aims at a critical reflection of our experience**s, and we discuss, what worked well or less well. Where possible, we give recommendations for future application projects. We divided the Discussion into two parts: a) the first part focusing more strongly on the MCDA process, and b) the second part following the transdisciplinary literature, again aiming to bridge between these two "worlds". We regard this as appropriate for a Special Issue focusing on Contributions of transdisciplinary approaches to hydrology and water resources management.

It is important to keep in mind that a one-off transdisciplinary project is unique, and the critical "quantitative" discussion is therefore strongly restricted. This is different from an **experimental design**, where the aim is a systematic hypothesis evaluation. We have done this in many other research projects, e.g., using online surveys (Aubert et al., 2020; Aubert &
* * *
[1] The word "co-design" was not appreciated by two referees, but we used it consistently within the FANFAR project, including all publications, and reportings to the EU. We regard it as appropriate and therefore adhere to it, see: https://fanfar.eu/about/. The main gist of "co-design" is to emphasize that a lot of effort was put into involving stakeholders in the process of building the flood forecasting system. This is rarely done, often it is rather a consulting approach that is employed in which stakeholders are presented with a final product towards the end of the project.

Lienert, 2019; Haag, Zuercher, et al., 2019; Lienert et al., 2016; Schmid et al., 2021; Zheng & Lienert, 2018). This is not the aim of the manuscript submitted to HESS.

It would be possible, to go into more detail, e.g., in the transdisciplinary Discussion section 4.2, but we did not do this for reasons of space. We are aware that the manuscript is long already. So the question is where should we place the focus? Should we expand the transdisciplinary discussion and critical reflection section 4.2 (e.g., with a table, which would, however be large)? Please see Table 3 in Lang et al. (2012).

**4e)** If we understand correctly, referee #1 seems to ask for a stronger **critical evaluation of the MCDA modelling parts**. This is of course also possible (and we have done that in other papers). We did provide some discussion in Section 4.1, which could be extended. We do not think that this is appropriate for this paper intended to be published in the SI on transdisciplinary projects, but kindly ask for some feedback on this point.
* * *
*5) The paper is about the design of a forecasting system and as such the topic is one that is common and well-studied in engineering. For an exemplary paper see: https://www.tandfonline.com/doi/full/10.1080/09544828.2016.1214693*

*Perhaps a note to this literature would be help the reader to place the article in the general literature.*

**Response:** Thank you for **suggesting this interesting paper**. However, we do not think that this paper is a strictly necessary citation for the submitted manuscript. It is – in our opinion – a technical engineering and MCDA paper (Unmanned Air System design), while ours under consideration here is about the transdisciplinary process (in a hydrology application). We cited many relevant MCDA textbooks and papers, many of them fundamental, e.g., Keeney (1982), with 318 citations. We do not see the need for this type of method literature from the engineering field.

**As a technical response**, the suggested paper uses AHP, a MCDA method (based on MAVT, but subject to considerable criticism) that we did not use in our approach. We used MAVT, as e.g., described in Keeney and Raiffa (1976). The proposed paper includes (interesting) game theory and optimization elements, which again we did not do. In our opinion, citing this paper would require additional methodological explanations to other possible approaches, which would increase confusion and lengthen the manuscript rather than adding to clarification. Why exactly this approach should be discussed, and not one of many other possible MCDA methods (e.g., outranking methods) is not obvious to us. MCDA comprises a family of methods and the field of Operational Research includes numerous possibilities of technical, mathematical, programming, and design approaches. We cannot see the benefit of entering this type of discussion, and especially not for a paper focusing on transdisciplinary processes.

Moreover, we found only minimal interaction with **real stakeholders** in this paper, documented in one sentence: "Two major stakeholders, user and manufacturer, were involved in a hybrid, cooperative/non-cooperative, non-zero sum, complete information, static game, modelling the interactions between their preferences and strategic choices, to accurately evaluate the alternative designs in the value-driven conceptual design of the UAS" (page 718). We therefore consider the aim of the paper to be completely different to ours under consideration.
* * *
*6) The paper strongly advocates the terms transdisciplinary and co-creation (in the text also co-coproduction, co-design). These are ideas which have been embedded in interactive MCDA for tens of years. MCDA uses data produced by experts from different fields. I personally do not see this as a novel co-creation or transdisciplinary process. In my opinion the paper reports a typical MCDA project and not anything else. There might be room for a separate paper discussing what new do the terms transdisciplinary and co-creation bring into MCDA and vice versa.*

**Response:** Please also see point 2), above. This manuscript was written in response to a call for papers for a SI on **transdisciplinary research** in hydrology. Regarding the use of the word "**co-design**", please see footnote 3, above.

Thank you for the observations. Indeed, the aim of the paper is to **focus on the transdisciplinary co-design process**. We are aware of the **MCDA** literature. Indeed, MCDA uses data produced by experts from different fields. This is the part described as "predictions" in the MCDA process, step 6 (Fig. 1). Additionally, we integrate the stakeholder preferences (step 5), which are – in our example – around 50 – 60 stakeholders participating in each of three workshops in West Africa. There is abundant MCDA literature available focusing on the integration and elicitation of stakeholder preferences; this is also not new. It is not about terminology: "transdisciplinary" is another field than "Operational Research", and within the latter "MCDA", "Behavioral Operational Research", or "Soft OR", all with a strong focus on interacting with "people in real-life experiences". We do not claim that we do something entirely new in the field of MCDA, and there have been other research projects from the transdisciplinary literature using MCDA. Moreover, there are different types of problems that can be approached with MCDA. One type concerns pre-defined problems (e.g., with a restricted number of decision makers, and with existing alternatives), where a method is used to solve the decision problem. Our decision concerns a complex problem, where a major part of the work is related to linking all those parts (e.g., clarifying the problem, defining the objectives, and building alternatives) to support decision-making. We aim to document the whole process, which is evidently scarce in the literature.

In this paper, we focus on making a transdisciplinary project accessible to hydrologists, with a relevant application example from hydrology, and in showing how this can be methodologically done using various problem structuring methods (termed "Soft OR" in the literature), and MCDA. In our view, **the innovation in the submitted manuscript** lies in the attempt to include the values and preferences of a large number of stakeholders from across a very large region (entire West Africa, 17 countries) in an iterative process (several consecutive workshops). This is **not a standard MCDA project, and it is not a standard hydrology project**. Moreover, high-quality MCDA applications in **developing countries** are scarce, to the best of our knowledge.

See point 4c, above. We would be willing to review this literature (MCDA applications in hydrology in Africa or in similar contexts), if the editors or referees consider this useful (?), and at the cost of increasing the length of the manuscript.
* * *
*7) The paper is full of poorly formulated sentences. I will not go into these in detail. As an example I have included a copy of the abstract with some exemplary points noted with question marks or BOLD text.*

**Response:** It would be helpful to have a few indications of what the referee means with "**poorly formulated sentences**". If needed, we can ask a professional corrector to proof read the manuscript.
* * *
*8) The abstract is way too long and anecdotal. Abstract. Climate change is projected to in-crease flood risks in West Africa. The EU Horizon 2020 project FANFAR co-designed a pre-operational flood forecasting and alert system for West Africa in three lively? workshops with 50–60 stakeholders, BY? adopting a transdisciplinary framework from Multi-Criteria Decision Analysis (MCDA). We aimed to (i) exemplify MCDA as a structured transdisciplinary process; (ii) prioritize suitable FANFAR system configurations; and (iii) document and discuss empiri-cal evidence WHAT IS THIS EVIDENCE DISCUSSED?. We used various interactive prob-lem structuring methods DID YOU REALLY USE MANY PROBLEM STRUCTURING METHODS OR SOME OTHER PROCEDURES  in stakeholder sessions to generate 10 ob-jectives and design 11 FANFAR system configurations. The non-additive MCDA model com-bined expert predictions about system performance with stakeholder preferences elicited in group sessions.A VERY STRANGE WAY OF SAYING THAT THE MCDA MODEL WAS BASED ON EXPERT DATA ON THE EXPECTED PERFORMANCE. All groups preferred a system producing accurate, clear, and accessible flood risk information that reaches recipi-ents well before floods. THIS WOULD BE THEIR IDEAL BUT PREFENCES RELALE TO TRADE-OFFS.To receive HOW IS THAT RECEIVED? this, most groups would trade off SPELLING, higher operation and maintenance costs, development time, and implementing several languages TRADE-OFF TO WHAT . We accounted for uncertainty in expert predic-tions with Monte Carlo simulation. Sensitivity analyses tested the results' robustness for changing MCDA aggregation models and diverging stakeholder preferences. Despite many uncertainties, three FANFAR system configurations achieved 63–70 % of the ideal case over all objectives in all stakeholder groups, and outperformed other options in cost-benefit visual-izations. VERY STRANGE CLAIM Stakeholders designed these best options to work relia-bly? under difficult West African conditions rather than incorporating many advanced features WHAT DOES THIS REFER TO?. The current OR THE PROPOSED?FANFAR system com-bines important features increasing system performance. Most respondents WHO? of a small online survey are satisfied, and willing to use the system in future THIS KIND OF SURVEY DOES NOT REALLY PROVE THAT THE SYSTEM WOULD BE USED IN REALITY- THE PAPER DID NOT CONSIDER ANY USABILITY QUESTIONS WHICH ARE ESSENTIAL WHEN DEVELOPING NEW SOFTWARE. We discuss our learning ? drawing from design principles of transdisciplinary research. We attempted to over-come CHECK SPELLING "unbalanced ownership" and "insufficient legitimacy" WHY THESE CONCERNS AND NOT E.G. LACK OF TRANSPARENCY OF THE MODELby including key West African institutions as consortium partners and carrying out co-design workshops with mandated rep-resentatives from 17 countries. MCDA overcomes TOO GENERAL STATEMENT challenges such as "lack of technical integration" WHAT DOES THIS MEAN AND HOW IS IT OVERCOME, or "vagueness and ambiguity of results". Whether FANFAR will have a "socie-tal impact" depends on long term financing and system uptake by West African institutions after termination of EU sponsoring. We hope ? that our promising results will have a "scien-tific impact" ON WHAT and motivate further DO YOU MEAN :STUDIES OF ?stakeholder en-gagement in hydrology research.*

**Response:** Thank you for these detailed inputs, which will allow us to **re-write the abstract**. However, there is a contradiction between the criticism that the abstract is too long, and re-quirements of methodological details. We cannot do both, and the methods are explained in the main body of the paper. We tried to summarize some main insights in the abstract, but

acknowledge that it was not well received. We will propose a revision of the abstract in the next stage and shorten the abstract.

Some detailed responses:

- (iii) document and discuss empirical evidence // WHAT IS THIS EVIDENCE DISCUSSED?.
  **Response:** Adding this increases the length of the abstract.

- We used various interactive problem structuring methods // DID YOU REALLY USE MANY PROBLEM STRUCTURING METHODS OR SOME OTHER PROCEDURES.
  **Response:** Indeed, as clearly outlined in the Methods Section, we used following **problem structuring methods**:
  * Section 2.3: Stakeholder analysis
  * Section 2.4: Generating objectives with: (a) an interactive online survey to first brainstorm, then select objectives from a master list; (b) same procedure in a pen and paper survey assisted by a moderator; (c) means-ends network in a moderated group discussion to come up with consensus objectives; and (d) plenary discussion and choice of the most important objectives by majority vote.
  * Section 2.4: Generating system options with: (a) "Strategy Generation Table"; (b) "Brainwriting 635" combined with "Cadavre Exquis"; (c) all FANFAR system options were discussed in a plenary session; and (d) as part of post processing, additional technically interesting options were created by FANFAR consortium members. We give extensive descriptions in the Supplementary Information for readers not familiar with these problem structuring methods.

- The non-additive MCDA model combined expert predictions about system performance with stakeholder preferences elicited in group sessions.A VERY STRANGE WAY OF SAYING THAT THE MCDA MODEL WAS BASED ON EXPERT DATA ON THE EXPECTED PERFORMANCE.
  **Response:** Thank you for indicating that the sentence is misleading. Correct is: "The MCDA model was based on **expert data** on the predicted performance **AND** on the **stakeholders' preferences**, which we elicited in group sessions." Please note the difference between expert predictions and stakeholder preferences; steps 5, 6, Fig. 1.

- All groups preferred a system producing accurate, clear, and accessible flood risk information that reaches recipients well before floods. THIS WOULD BE THEIR IDEAL BUT PREFENCES RELALE TO TRADE-OFFS.
  **Response:** How we **elicited the trade-offs** between achieving objectives is described in detail in the main text (including lengthy descriptions for readers not familiar with MCDA methods in the Supplementary Information). Please see Section 2.7.2 Weight: We used the Swing method and an adaptation of Simos' revised card procedure. We do not think that such method details belong into the abstract.

- To receive HOW IS THAT RECEIVED? this, most groups would trade off SPELLING, higher operation and maintenance costs, development time, and implementing several languages TRADE-OFF TO WHAT .
  **Response:**
  * We use **MCDA** to receive results about the best performing system.
  * Spelling: **HESS recommends using hyphens sparingly**. Therefore, we wrote "trade off" without hyphens. If the editors prefer "trade-off" with hyphen, we are happy to do so.
  * Thank you for pointing out that the **trade-off between objectives** is unclear. To the stakeholders it was more important to have a system that produces accurate, clear, and accessible flood risk information that reaches recipients well before floods, and

they were willing to accept higher costs, higher development time, and fewer languages.

- ..achieved 63–70 % of the ideal case over all objectives in all stakeholder groups, and outperformed other options in cost-benefit visualizations. VERY STRANGE CLAIM
**Response:** this claim is based on **cost-benefit visualizations** of the MCDA results; see Fig. 7 in the main body of the paper.

- *..than incorporating many advanced features WHAT DOES THIS REFER TO?.*
**Response:** Again, we would have to increase the length of the abstract by including these details. The **options that performed best in the MCDA were not those options with the most advanced features** such as "earth observations and in situ data" for hydrological observations and meteorological forcing.

- *The current OR THE PROPOSED?FANFAR system combines important features increasing system performance.*
**Response:** the "**current FANFAR system**" is correct, because there is already a system available. See: https://fanfar.eu/ivp/

- Most respondents WHO? of a small online survey are satisfied, and willing to use the system in future THIS KIND OF SURVEY DOES NOT REALLY PROVE THAT THE SYSTEM WOULD BE USED IN REALITY- THE PAPER DID NOT CONSIDER ANY USABILITY QUESTIONS WHICH ARE ESSENTIAL WHEN DEVELOPING NEW SOFTWARE.
**Response:**
* The **respondents** were stakeholders (included in the previous workshops) that participated in a fourth workshop, which had to be carried out online in January 2021 (due to the COVID pandemic). See methods.
* We fully agree that **positive feedback does not mean that the system will be used** in future. This problem arises in any transdisciplinary project, and we shortly discussed it in Section 4.2.3. (Re-)integrating and applying the co-produced knowledge.
* The **(written) technical usability feedback** is not at the core of the current paper. We explain this in section 2.2 (page 6): "At each workshop, West African represent-atives presented the local flood situation in their country during rainy seasons and their experience with using the FANFAR system. Each workshop hosted extensive technical sessions for experimentation with the latest FANFAR system, including structured technical feedback. Between workshops, the pre-operational system was adapted to meet requests as well as possible (Andersson, Ali, et al., 2020; Andersson, Santos, et al., 2020). In this paper, we focus only on interactions that are at the core of the MCDA process."

- We discuss our learning ? drawing from design principles of transdisciplinary research. We attempted to over-come CHECK SPELLING "unbalanced ownership" and "insufficient legitimacy" WHY THESE CONCERNS AND NOT E.G. LACK OF TRANSPARENCY OF THE MODELby including key West African institutions as consortium partners and carrying out co-design workshops with mandated representatives from 17 countries. MCDA overcomes TOO GENERAL STATEMENT challenges such as "lack of technical integration" WHAT DOES THIS MEAN AND HOW IS IT OVERCOME, or "vagueness and ambiguity of results". Whether FANFAR will have a "societal impact" depends on long term financing and system uptake by West African institutions after termination of EU sponsoring. We hope ? that our promising results will have a "scientific impact" ON WHAT and motivate further DO YOU MEAN :STUDIES OF ?stakeholder engagement in hydrology research.
**Response:** Thank you for this feedback, which indicates that we did not well convey our message. **We will revise this in a next version**. This section in the abstract is

based on the design principles of transdisciplinary research processes in (Lang et al., 2012), and identified challenges (see Table 3 in that paper). We use this approach to discuss our learning and insights gained in the FANFAR project; see section 4.2: Challenges of transdisciplinary process and lessons learnt.
* * *
*Some more remarks:*

*9) The paper uses the terms expert predictions, predicted outcome (section 2.8) and data ( Box 7 in Fig.1.) . Why not Uuse the word data always?*

**Response: "Data"** is a general term. To be specific, we use the term "expert predictions", "predicted outcome" step 6 in Fig. 1, and "preferences" step 5 in Fig. 1. BOTH are data that enter the MCDA model in step 7. This is standard MCDA terminology. We differentiate between those terms, due to their distinct nature. One relates to the subjective nature of the data, the preferences, the other to the knowledge in a specific field, the predictions. Thus, the general term for step 6 is "predictions", which can be derived from models, the literature, or experts ("expert predictions"), depending on the case.
* * *
*10) Section 2.8: The authors make strong claims in favor of non-linear models but do not critically discuss the new problems created. For example, the introduction of the coefficient gamma is not at all simple. How do you justify and explain the value selected for gamma to the stakeholders? The justification that attributes are non-compensatory can also be challeged. Typically any system design starts with some minimum requirement with respect to the attributes and only when these are met one starts to compare extra features.*

**Response:** We use the **non-additive model** exactly because the stakeholders did not agree with the implications of this standard model, as explained in section 2.7.2 (page 10): "To check for the validity of the additive aggregation model (sect. 2.8), we shortly discussed implications in the weight elicitation sessions using elicitation procedures from our earlier work (Haag, Lienert, et al., 2019; Zheng et al., 2016)." Referee #1 is right that we did not emphasize this disagreement of stakeholders with model implications in the Results and will add it.

Additionally, we did indeed carry out "system design" with **different aggregation models**, including the additive model, but also other aggregation models with different levels of Gamma. Please see results of the **sensitivity analyses** in Section 3.7, and Table 2 (page 20, 21). Note that our approach is different from the engineering system design approach of referee #1. We do not start with the "minimum system requirement", but with the MCDA model that best fits the stakeholders' preferences, which we elicited from the stakeholders in workshop sessions. We challenged the implications of using this model by changing model parameters in the sensitivity analyses.

We highlight that MCDA, coupled with other methods (e.g., problem structuring from the field of "Soft OR"), does not provide the answer by itself, but can effectively support decision-making. Acknowledging that, the use of non-additive models allows us to provide additional details to the decision-makers at a cost of difficult elicitation issues (e.g., gamma values). Thus, if the analysts are able to understand whether the additive model holds or not, they should

prioritize the assessment of non-additive models in sensitivity analysis, where they can assess its impact without incurring in those difficult challenges (e.g., gamma value elicitation). This means that we cannot provide the exact non-additive model without tackling those problems (e.g., gamma value elicitation), but we can provide additional information able to support decision-making in complex situations, which is the initial requirement. The nature of MCDA, from its different strands, points exactly in that direction. As examples for this discourse please see differentiation between "two clearly different conceptions on which decision aiding can be carried out. (…) primarily positivist, and (…) primarily constructivist" (Roy, 2010). Moreover: "(…) structuring decision problems is the most important and, at the same time, least well understood task of a decision analyst" (von Winterfeldt & Edwards, 2007).
* * *
*11) Also the aggregation of opinions by taking averages is quite problematic as the result is then nobody´s opinion.*

**Response:** We fully agree, which is why we did not do this! We **elicited the weights** in different groups and treated each group individually. Here, we used the average within each group. However, we additionally elicited and recorded outlier (extreme) opinions within each group, and tested the implications of these outliers with the **sensitivity analyses**. Thus, we captured the entire range of possible preferences. We went a step further and tested "possible even extremer" preferences by doubling the weight of the objective "Several languages", because we thought that its importance might have been underestimated by the stakeholders (based on general discussions in the workshops). All these sensitivity analyses had little effect on changing the ranking of the best performing options. Please see Table 2.
* * *
*12) The notation and names used for the options and variables etc. are so difficult that the reader really cannot follow the text. Are they all needed? Who can understand Table 1 and 2.*

**Response:** We think that **Table 1** provides a useful overview for hydrologists interested in the more technical features of the flood forecast and alert system. The variables in Table 1 should be understandable to hydrologists working in the field. We agree that the short names of options (ID, Tab. 1) might be difficult to understand intuitively, but increasing the length of these names makes Figures and Tables unreadable.

We do hope that readers of HESS understand a correlation matrix (Table 2) based on the nonparametric **Kendall's τ correlation coefficient** (Kendall, 1938) to measure rank reversals (as in e.g., Zheng et al., 2016). We define and explain the different sensitivity analyses (column 2 in Table 2) in detail in the Methods section 2.9.2 Sensitivity analyses of aggregation model and stakeholder preferences (page 11, 12).
* * *
*13) Discussion is not a discussion of the approach but a report of the process. The claim that this co-design would meet the main requirements is strong. The discussion of the process continues in an anecdotal mode in 4.1.1. Where is critical evaluation?*

**Response:** Please see our response to point 4, above.
* * *
*14) 4.1.2. is quite strange and unclear and full of unsupported claims with these cryptic notations. Quote: The weights indicate that most groups preferred that the system produces accurate, clear, and reliable information that reaches recipients well before a flood (11_accur_info; 12_clear_info; 21_reliable_info; 22_timely info; Figure 4)*

**Response:** this sentence relates to the objectives (see Fig. 2): "accurate, clear, and reliable information that reaches recipients well before a flood". In parentheses, we give the according **short names**, which were used in Figure 4. We agree that they are cryptic, but longer names are not usable for the Figures. We think that Fig. 4 visualizes which objectives the different stakeholder groups consider as more (or less) important.
* * *
*15) 4.2.2. This section uses concepts and statements from Lang et al. but in this context they remain un-understandable such as this: `` Possible coping strategies are "Low thresholds for and appropriate levels of participation" (Lang et al., 2012)``*

**Response:** We agree. For reasons of length, we did not **elaborate on the concepts in Lang et al. (2012).** Please see point 4d) above; we kindly ask the editors for clarification, whether we should increase the length of the manuscript here.
* * *
16) *In this section the authors seem to accept a preference modelling approach which is not understandable to the stakeholders. This is not really supported by the professionals in the field.*

**Response:** Which section does referee #1 refer to? **Who are the "professionals**", and in "which field"? Does it refer to section 4.1.2 Dealing with uncertainty of predictions, preferences, and model assumptions, and the discussion on the importance of stakeholders understanding the model? If yes, we cite Raimo Hämäläinen (2015), who has received prestigious awards; e.g., "the Ramsey Medal of INFORMS in 2019 for his distinguished contributions in decision analysis". To quote: «The Ramsey medal is the highest award that the DAS (Decision Analysis Society) can bestow upon its members, and recognizes the breadth and impact of professor Hämäläinens career. His research covers models for decision making ranging from negotiation support to pilot decision making and participatory multi-criteria methods for environmental policy." He is certainly one of the most prominent experts in MCDA, Decision Analysis, Modelling with stakeholders, Systems analysis, and Behavioral Operational Research, see https://sal.aalto.fi/en/personnel/raimo.hamalainen/.

Note: we use an opposite approach than what referee #1 suggests: instead of using a (simpler) model with the benefit of being able to explain fully the math to the stakeholders, we elicit the preferences from stakeholders regarding the model implications. In our case: does the additive aggregation model meet their expectations or should the model allow for interaction between objectives and allow for partial compensation between objectives? The stakeholders understood this, when we discussed it with them in the workshops, which is why we chose a non-additive model as default. However, to be on the safe side, we tested possible effects on the MCDA results in sensitivity analyses using e.g., the additive aggregation model; see point 10) above. We wrote (page 28): "In a recent paper about behavioral issues

in modelling, Hämäläinen (2015; Table 2) recommends using "transparent and simplified models for learning (when modelling together with stakeholders), and comprehensive models for problems solving". For FANFAR, local sensitivity analyses sufficed to conclude that additive aggregation has an effect, but does not alter the ranking of the best performing options. In other words, the results of the sensitivity analyses are relevant additional information for the decision-makers.
* * *
*17) The conclusion that system would be used in reality, when people say that they intend to use it, not really supported real life experience. There is no guarantee of real use even if people say they will use . User acceptance is a much more complicated issue and depends on the system and interface design and not only on the configuration. There is a rich literature on human computer interaction and the user acceptance of software*

**Response:** We agree, see point 8) above w.r.t. stakeholder feedback. We did not write, "People will use it".

We write (page 28): "The **online survey allowed receiving some feedback** about how well the current FANFAR system meets the 10 objectives, despite not being able to carry out a workshop in Africa due to the COVID-19 pandemic. The stakeholders were quite satisfied with its performance during the 2020 rainy season (Figure 8), and are generally **willing** to use the FANFAR system in future."

Page 30: "A final challenge is *"Tracking of scientific and societal impacts"*, as standardized approaches to evaluate the outcomes of transdisciplinary research are still missing. This requires follow-up studies, which are integrated in current proposals for full operationalization of the FANFAR system. AGRHYMET, mandated by ECOWAS to provide flood forecast and alert information across West Africa, has the authority to drive the uptake and operationalization of the FANFAR system. The project included several components to facilitate long-term sustainability (e.g., co-development, open source tools, documentation, training material, and capacity development activities). It is encouraging that AGRHYMET already uses the FANFAR system beyond project activities (e.g., at the PRESASS and PRESAGG forums (WMO, 2021), to support the ECOWAS flood management strategy, and in their MSc curriculum). Nevertheless, long-term sustainability and operationalization after termination of EU sponsoring is still not secured. AGRHYMET now leads the effort to secure financing for the FANFAR system, along with other consortium partners. As this challenge is not listed in Lang et al. (2012), we propose adding it to the framework."

**References**

Andersson, J., Ali, A., Arheimer, B., Crochemore, L., Gbobaniyi, B., Gustafsson, D., Hamatan, M., Kuller, M., Lienert, J., Machefer, M., Magashi, U., Mathot, E., Minoungou, B., Naranjo, A., Ndayizigiye, T., Pacini, F., Silva Pinto, F., Santos, L., Shuaib, A., & , , 2020 (2020). *Flood forecasting and alerts in West Africa − experiences from co-developing a pre-operational system at regional scale* EGU General Assembly 2020, 4–8 May 2020, Online. https://doi.org/10.5194/egusphere-egu2020-7660

Andersson, J., Santos, L., Isberg, K., Gustafsson, D., Musuuza, J., Minoungou, B., Crochemore, L., & :, R. f. a. a. (2020). *Deliverable: D3.2. Report documenting and explaining the hydrological models* [Deliverable](D3.2). F. Consortium. https://fanfar.eu/resources/

Aubert, A. H., Esculier, F., & Lienert, J. (2020). Recommendations for online elicitation of swing weights from citizens in environmental decision-making. *Operations Research Perspectives, 7,* Article 100156. https://doi.org/10.1016/j.orp.2020.100156

Aubert, A. H., & Lienert, J. (2019). Gamified online survey to elicit citizens' preferences and enhance learning for environmental decisions. *Environmental Modelling & Software, 111,* 1-12. https://doi.org/10.1016/j.envsoft.2018.09.013

Bond, S. D., Carlson, K. A., & Keeney, R. L. (2008). Generating objectives: Can decision makers articulate what they want? *Management Science, 54*(1), 56-70. <Go to ISI>://000252420600005

Eisenführ, F., Weber, M., & Langer, T. (2010). *Rational Decision Making* (Vol. 1st ed.). Springer.

Haag, F., Lienert, J., Schuwirth, N., & Reichert, P. (2019). Identifying non-additive multi-attribute value functions based on uncertain indifference statements. *Omega-International Journal of Management Science, 85,* 49-67. https://doi.org/10.1016/j.omega.2018.05.011

Haag, F., Zuercher, S., & Lienert, J. (2019). Enhancing the elicitation of diverse decision objectives for public planning. *European Journal of Operational Research*, *279*(3), 912-928. https://doi.org/10.1016/j.ejor.2019.06.002

Hämäläinen, R. P. (2015). Behavioural issues in environmental modelling – The missing perspective. *Environmental Modelling & Software*, *73*, 244-253. https://doi.org/http://dx.doi.org/10.1016/j.envsoft.2015.08.019

Keeney, R. L. (1982). Decision-Analysis - an Overview. *Operations Research*, *30*(5), 803-838. https://doi.org/https://pubsonline.informs.org/doi/abs/10.1287/opre.30.5.803

Keeney, R. L., & Raiffa, H. (1976). *Decisions with Multiple Objectives: Preferences and Value Tradeoffs*. Wiley.

Kendall, M. G. (1938). A new measure of rank correlation. *Biometrika*, *30*, 81-93. https://doi.org/10.2307/2332226

Lang, D. J., Wiek, A., Bergmann, M., Stauffacher, M., Martens, P., Moll, P., Swilling, M., & Thomas, C., J. (2012). Transdisciplinary research in sustainability science: practice, principles, and challenges. *Sustainability Science*, *7*(Supplement 1), 25-43.

Lienert, J., Duygan, M., & Zheng, J. (2016). Preference stability over time with multiple elicitation methods to support wastewater infrastructure decision-making. *European Journal of Operational Research*, *253*, 746-760. https://doi.org/http://dx.doi.org/10.1016/j.ejor.2016.03.010.

Lienert, J., Schnetzer, F., & Ingold, K. (2013). Stakeholder analysis combined with social network analysis provides fine-grained insights into water infrastructure planning processes. *Journal of Environmental Management*, *125*, 134-148. https://doi.org/10.1016/j.jenvman.2013.03.052

Reichert, P., Niederberger, K., Rey, P., Helg, U., & Haertel-Borer, S. (2019). The need for unconventional value aggregation techniques: experiences from eliciting stakeholder preferences in environmental management [journal article]. *EURO Journal on Decision Processes*, *7*(3), 197-219. https://doi.org/10.1007/s40070-019-00101-9

Roy, B. (2010). Two conceptions of decision aiding [Article]. *International Journal of Multicriteria Decision Making*, *1*(1), 74-79. https://doi.org/10.1504/IJMCDM.2010.033687

Schmid, S., Vetschera, R., & Lienert, J. (2021). Testing Fairness Principles for Public Environmental Infrastructure Decisions. *Group Decision and Negotiation*, *30*, 611-640. https://doi.org/10.1007/s10726-021-09725-2

von Winterfeldt, D., & Edwards, W. (2007). Defining a decision analytic structure. In W. Edwards, J. Miles, R.F., & D. von Winterfeldt (Eds.), *Advances in decision analysis: From foundations to applications, chapter 6* (pp. 81-103). Cambridge University Press. https://scholar.google.com/scholar?hl=en&as_sdt=0%2C5&q=von+Winterfeldt%2C+D.%2C+Edwards%2C+W.%2C+2007%2C+Defining+a+decision+analytic+structure&btnG=

WMO. (2021). *WMO (World Meteorological Organization): Regional Climate Outlook Forums*. WMO (World Meteorological Organization). Retrieved 25.02.2021 from https://public.wmo.int/en/our-mandate/climate/regional-climate-outlook-products

Zheng, J., Egger, C., & Lienert, J. (2016). A scenario-based MCDA framework for wastewater infrastructure planning under uncertainty. *Journal of Environmental Management*, *183, Part 3*, 895-908. https://doi.org/http://dx.doi.org/10.1016/j.jenvman.2016.09.027

Zheng, J., & Lienert, J. (2018). Stakeholder interviews with two MAVT preference elicitation philosophies in a Swiss water infrastructure decision: Aggregation using SWING-weighting and disaggregation using UTAGMS. *European Journal of Operational Research*, *267*(1), 273-287. https://doi.org/https://doi.org/10.1016/j.ejor.2017.11.018

---

## Author Comment (AC2)

Eawag
Überlandstrasse 133
P.O. Box 611
8600 Dübendorf
Switzerland
+41 (0)58 765 55 44
www.eawag.ch

Environmental Social Sciences (ESS)
Dr. Judit Lienert
Cluster Leader Decision Analysis
+41 (0)58 765 55 74
judit.lienert@eawag.ch
https://www.eawag.ch/en/aboutus/portrait/organisation/staff/profile/judit-lienert/show/

[Figure]

Dübendorf, 23 June 2021

**Response to referee # 2**

Dear Referee

Thank you very much for reviewing our manuscript:

**Judit Lienert, Jafet Andersson, Daniel Hofmann, Francisco Silva Pinto, Martijn Kuller, "Using Multi-Criteria Decision Analysis for transdisciplinary co-design of the FANFAR flood forecasting and alert system in West Africa". hess-2021-177**

This manuscript was written for the HESS Special Issue **"Contributions of transdisciplinary approaches to hydrology and water resources management"**

We are grateful for the work that has gone into reviewing our paper. We do know that this takes a lot of time, which receives no direct reward. We are very willing to improve the manuscript based on your inputs, wherever possible.

We have addressed your comments one-by-one below. *The referees' comments are given in Italics*, our response is given in normal font.

We look forward to suggestions for improving the manuscript so that it meets requirements of publications in HESS.

With best regards,

Judit Lienert

also on behalf of my co-authors, Jafet Andersson, Daniel Hofmann, Francisco Silva Pinto, and Martijn Kuller

*General comments:*

*1) While the paper does address relevant scientific questions within the scope of Hydrology and Earth System Sciences, it does not do so in a novel, innovative and comprehensible way. The manuscript in its current format seems to resemble a project report in memo style that describes what has been done in the FANFAR project.*

**Response:**

**1a) Report of transdisciplinary process: Same response as to referee #1, point 1.** We do see that the manuscript can be read as "a project report in memo style". Our aim was to address the call of the Special Issue on Contributions of transdisciplinary approaches to hydrology and water resources management, for which the paper was written: "While interdisciplinary conversations have been happening to some extent, **transdisciplinary endeavours remain largely undocumented**. The type of transdisciplinarity we are interested in here engages, broadly speaking, academic and non-academic perspectives in knowledge production." (See the call of the SI)

**1b) Comprehensibility and structure: Same response as to referee #1, point 4a). Our aims** are presented in section 1.1: "(i) to exemplify the use of Multi-Criteria Decision Analysis (MCDA) as methodological framework for integrating stakeholders in a structured co-design process; (ii) to prioritize development of suitable FANFAR flood forecast and alert system configurations based on expert estimates about system performance as well as stakeholder preferences; and (iii) to document empirical evidence of a large transdisciplinary, transcontinental co-design process, and discuss insights, lessons learnt, and recommendations of special interest to hydrology praxis when engaging with stakeholders and society." We think that the aims match the call of the SI.

If a problem of the manuscript is that our research aims are not formulated as **research questions**, we propose to re-formulate the aims as specific research questions, and would **re-structure the paper** to answer these, e.g. along these lines:

- (i) **What would characterize a good regional-scale flood forecasting and alert system for West Africa?** In other words, how do the MCDA results contribute to finding a suitable FANFAR system configuration? Is it possible to find a "good compromise system configuration", despite large uncertainty (of expert predictions about FANFAR system performance, and of MCDA model), and despite the fact that stakeholders may have strongly different preferences about what the system should look like?
- (ii) **How can a large number of stakeholders be integrated** into a transdisciplinary process aimed at designing a flood forecast and alert system for entire West Africa (FANFAR system)?
- (iii) How much do the early problem structuring steps help **focusing the development of the FANFAR system at the beginning of the project to meet the stakeholders' expectations** (i.e., before the MCDA results are available)? How well does this early focus match later MCDA results?
- (iv) **What worked well, what worked less well?** What insights, lessons learnt, and recommendations can we provide to hydrology praxis when engaging with stakeholders and society?

We kindly ask for feedback (e.g., by the editors), whether this restructuring and reformulation into research questions would be helpful?

**1c) Innovation: Same response as to referee #1, point 4c (also point 1a, above).** Additionally, we contribute to documenting transdisciplinary research projects. We are not aware of many transdisciplinary projects that carry out a thorough MCDA process with that many stakeholders, from 17 countries, in several large workshops (50 – 60 participants), in hydrology, in Africa (or in similar contexts). Hence, we consider this application to be an innovation in itself. However, we have not reviewed this literature. We would be willing to do so, if the editors or referees consider this type of review useful. We do wish to point out, that such additions will again add to the length of the manuscript.

**Same response as to referee #1, point 6.** In this paper, we focus on making a transdisciplinary project accessible to hydrologists, with a relevant application example from hydrology, and in showing how this can be methodologically done using various problem structuring methods (termed "Soft OR" in the literature), and MCDA. In our view, the innovation in the submitted manuscript lies in the attempt to include the values and preferences of a large number of stakeholders from across a very large region (entire West Africa, 17 countries) in an iterative process (several consecutive workshops). This is not a standard MCDA project, and it is not a standard hydrology project. Moreover, high-quality MCDA applications in developing countries are scarce, to the best of our knowledge.

See point 4c (referee #1). We would be willing to review this literature (MCDA applications in hydrology in Africa or in similar contexts), if the editors or referees consider this useful (?), and at the cost of increasing the length of the manuscript.
* * *
*2) Already the very long abstract leaves readers confused about the actual research question, the target audience, the methodological innovation, the novel results and derived insights.*

**Response. Abstract, same response as to referee #1, point 8.** Thank you. We will propose a revision of the abstract in the next stage and shorten the abstract.
* * *
*3) Not much more clarity can be gained from reading the full lengthy report, which could not only be shortened and streamlined but also better structured and more clearly written (native speaker check needed; jargon and buzzword heavy) to qualify as a journal publication.*

**Response:**

**3a) Clarity, shortening**; see point 1b, above. We will restructure the manuscript, after having received some indication on where the focus should be. However, it will not be possible to meet all the raised additional requirements (see referee #1 in various places), and point 1c, above, and to shorten the manuscript. We can delete parts of the discussion or any other parts that the editors consider appropriate, but would appreciate feedback on which parts. This would weaken the documentation and critical discussion of the transdisciplinary process, or the integration into the current MCDA and transdisciplinary literature (or both).

**3b) Native speaker check, jargon and buzzword heavy; same response as to referee #1, point 7).** If needed, we can ask a professional corrector to proof read the manuscript. Should we do this? Moreover, we will delete some of the more biased assumptions and jargon in the revisions (see referee #3, point 3).
* * *
*Specific comments:*

*4) I understand that the authors argue that their MCDA approach is more transdisciplinary in nature - emphasized by adding the buzzword co-design - than existing MCDA approaches. Reading the methods, results and discussion section I do, however, not see this claim substantiated. If the authors still see this as the USP of their contribution I suggest that their work must be better embedded in and contrasted with the existing MCDA literature. Maybe there really is a methodological innovation that has scientific and policy relevance - in the current manuscript this 'treasure' is very effectively hidden though (see general comments on the overall quality of this preprint above).*

**Response:**

**4a) Co-design:** The word "co-design" was not appreciated by two referees, but we used it consistently within the FANFAR project, including all publications, and reportings to the EU. We regard it as appropriate and therefore adhere to it, see: https://fanfar.eu/about/. The main gist of "co-design" is to emphasize that a lot of effort was put into involving stakeholders in the process of building the flood forecasting system. This is rarely done, often it is rather a consulting approach that is employed in which stakeholders are presented with a final product towards the end of the project.

**4b) MCDA and transdisciplinarity.** Thank you for clarifying. We wish to emphasize that we do not argue that our "MCDA approach is more transdisciplinary than existing MCDA approaches." We are not aware of making this statement. Rather, we aim "(i) to exemplify the use of Multi-Criteria Decision Analysis (MCDA) as methodological framework for integrating stakeholders in a structured co-design process", i.e., show how MCDA can support a transdisciplinary process. We try to bridge between the transdisciplinary literature and the MCDA literature. Moreover, our contribution is about making hydrological forecasting more transdisciplinary. As mentioned in the call of the Special Issue, our work can also contribute to "transnational knowledge exchange regarding impacts of different transdisciplinary projects in different contexts."

**Same as response 6 to referee #1. Response:** Thank you for the observations. Indeed, the aim of the paper is to focus on the transdisciplinary co-design process. We are aware of the MCDA literature. Indeed, MCDA uses data produced by experts from different fields. This is the part described as "predictions" in the MCDA process, step 6 (Fig. 1). Additionally, we integrate the stakeholder preferences (step 5), which are – in our example – around 50 – 60 stakeholders participating in each of three workshops in West Africa. There is abundant MCDA literature available focusing on the integration and elicitation of stakeholder preferences; this is also not new. It is not about terminology: "transdisciplinary" is another field than "Operational Research", and within the latter "MCDA", "Behavioral Operational Research", or "Soft OR", all with a strong focus on interacting with "people in real-life experiences". We do not claim that we do something entirely new in the field of MCDA, and there have been other research projects from the transdisciplinary literature using MCDA. Moreover, there are different types of problems that can be approached with MCDA. One type concerns pre-defined

problems (e.g., with a restricted number of decision makers, and with existing alternatives), where a method is used to solve the decision problem. Our decision concerns a complex problem, where a major part of the work is related to linking all those parts (e.g., clarifying the problem, defining the objectives, and building alternatives) to support decision-making. We aim to document the whole process, which is evidently scarce in the literature.

In this paper, we focus on making a transdisciplinary project accessible to hydrologists, with a relevant application example from hydrology, and in showing how this can be methodologically done using various problem structuring methods (termed "Soft OR" in the literature), and MCDA. In our view, the innovation in the submitted manuscript lies in the attempt to include the values and preferences of a large number of stakeholders from across a very large region (entire West Africa, 17 countries) in an iterative process (several consecutive workshops). This is not a standard MCDA project, and it is not a standard hydrology project. Moreover, high-quality MCDA applications in developing countries are scarce, to the best of our knowledge.

See point 1c, above. We would be willing to review this literature (MCDA applications in hydrology in Africa or in similar contexts), if the editors or referees consider this useful (?), and at the cost of increasing the length of the manuscript.

---

## Author Comment (AC3)

Eawag
Überlandstrasse 133
P.O. Box 611
8600 Dübendorf
Switzerland
+41 (0)58 765 55 44
www.eawag.ch

Environmental Social Sciences (ESS)
Dr. Judit Lienert
Cluster Leader Decision Analysis
+41 (0)58 765 55 74
judit.lienert@eawag.ch
https://www.eawag.ch/en/aboutus/portrait/organisation/staff/profile/judit-lienert/show/

[Figure]

Dübendorf, 23 June 2021

**Response to referee # 3**

Dear Referee

Thank you very much for reviewing our manuscript:

**Judit Lienert, Jafet Andersson, Daniel Hofmann, Francisco Silva Pinto, Martijn Kuller, "Using Multi-Criteria Decision Analysis for transdisciplinary co-design of the FANFAR flood forecasting and alert system in West Africa". hess-2021-177**

This manuscript was written for the HESS Special Issue **"Contributions of transdisciplinary approaches to hydrology and water resources management"**

We are grateful for the work that has gone into reviewing our paper. We do know that this takes a lot of time, which receives no direct reward. We are very willing to improve the manuscript based on your inputs, wherever possible.

We have addressed your comments one-by-one below. *The referees' comments are given in Italics*, our response is given in normal font.

We look forward to suggestions for improving the manuscript so that it meets requirements of publications in HESS.

With best regards,

Judit Lienert

also on behalf of my co-authors, Jafet Andersson, Daniel Hofmann, Francisco Silva Pinto, and Martijn Kuller

**Anonymous Referee #3**

*1) The paper reports on the FANFAR project and its opportunities and challenges of developing a reliable and useable flood forecast system in West and Central Africa. It discusses the opportunities and challenges of integrating stakeholder knowledge and producing both scientific reliable and useful information (more about this in Lemos and Morehouse 2005). In section 1.3 MCDA is presented as a remedy for all transdisciplinary projects supported with the enumeration of six central lines of argumentation. In sum, the paper is rather descriptive than analytic. However, this is a very common problem of reporting about transdisciplinary projects, which is depended on transparent and thick descriptions about how processes of knowledge integration haven been implemented, how "data" from non-scientific experts is included and so on. The paper handles this endeavor with sufficient accuracy. In addition, a comprehensive annex is provided with helpful tables and feedback derived from stakeholder surveys.*

*The paper has a lot of merits of getting published, but I have also three critical comments:*

**Response:** Thank you for this **overall positive evaluation**. We agree that the paper is rather descriptive than analytic, which is, as referee #3 states, a common problem of reporting about transdisciplinary projects. We are grateful that the referee acknowledges that we handle the endeavor of "knowledge integration" with sufficient accuracy.

Thank you for suggesting the interesting paper, which is indeed an important contribution. Already in the abstract, it states that: "It finds that although no single model can fulfill the multitude of goals of such assessments, **it is in highly interactive models** that the possibilities of higher levels of **innovation and related social impact are most likely to occur**" (Lemos & Morehouse, 2005). This follows the discourse that we shortly touched upon in the last part of the Discussion section 4.1.2. ("… , and model assumptions"), namely the paper by Hämäläinen (2015). There would be much more to add to this discussion about **modelling with and for stakeholders** (see e.g., Voinov et al., 2016). It is a question of space and priority, how strongly we can and should expand this topic.

It was not our intention to **advocate "MCDA (as …) a remedy** for all transdisciplinary projects". We will do our best to correct this impression in the revisions.
* * *
*2) The paper starts by defining transdisciplinarity using a for the context very appropriate definition of Lang et al. 2012. However, the discussion about the aims and obstacles of transdisciplinary research is almost exclusively referencing this source (and not others). This is far away of being a comprehensive literature review adequate for a journal publication. In addition, please explain (briefly) the "ordinary" challenges of transdisciplinary research in a transnational context already at the beginning and not only in the discussion.*

**Response:** Thank you, we again agree. We had written a longer **introduction part on transdisciplinary research,** but had deleted it for reasons of space. We are fine with including additional references and expanding this part somewhat. However, rather than including a comprehensive review on transdisciplinary research, we suggest focusing on main papers from the transdisciplinary field, followed by a review about concrete transdisciplinary projects using MCDA in hydrology, in Africa. This follows the request of referee #1 (point 4c, point 6) and referee #2 (point 1c). We kindly ask for some guidance by the referee and editors: do

you agree with this prioritization, and should we increase the length of the paper with a litera-ture review at all?

We will shortly clarify what we meant with "ordinary" challenges of transdisciplinary projects, or delete if inappropriate for the revised manuscript.
* * *
*3) I would suggest **reducing some of the more biased assumptions** like "lively work-shops", "FANFAR project is unique", "unique practice and outcome oriented project", "pro-ducing a good flood forecast and alert system" … to prevent the impression of reading a pro-ject proposal or advertisement and not a scientific paper. If you want to judge your own pro-ject, you would have better stick to an evaluation of the project by other researchers or at least to a survey among participants.*

**Response:** We are willing to do so.
* * *
*4) My third point questions parts of the structure. In the methods section **I would suggest focusing on methods and tools of conducting and writing the paper** and not on how the FANFAR project and its transdisciplinary methods were implemented. I would rather add an-other main section called "**Processes of transdisciplinarity**" (or something similar), where the main project's undertakings are described.*

**Response:** We do not understand what is meant with focusing on "conducting and writing the paper". We think it is important to describe the methods used in interaction with the stakeholders. Describing each step of the problem structuring and MCDA process increases the clarity in our opinion, also to those less familiar with transdisciplinary approaches, and MCDA in particular. However, one idea in response to referees #1 (point 4a) and #2 (point 1b) is to **focus more strongly on research questions**, and re-structure the paper accord-ingly. We kindly ask the referee and editors for advice on this.

**References**

Hämäläinen, R. P. (2015). Behavioural issues in environmental modelling – The missing perspective. *Environmental Modelling & Software*, *73*, 244-253. https://doi.org/http://dx.doi.org/10.1016/j.envsoft.2015.08.019

Lemos, M. C., & Morehouse, B. J. (2005). The co-production of science and policy in integrated climate assessments. *Global Environmental Change-Human and Policy Dimensions*, *15*(1), 57-68. https://doi.org/10.1016/j.gloenvcha.2004.09.004

Voinov, A., Kolagani, N., McCall, M. K., Glynn, P. D., Kragt, M. E., Ostermann, F. O., Pierce, S. A., & Ramu, P. (2016). Modelling with stakeholders – Next generation. *Environmental Modelling & Software*, *77*, 196-220. https://doi.org/http://dx.doi.org/10.1016/j.envsoft.2015.11.016